



# A North Sea in situ evaluation of the Fitch Wind Farm Parametrization within the Mellor–Yamada–Nakanishi–Niino and 3D Planetary Boundary Layer schemes

Nathan J. Agarwal[1], Julie K. Lundquist[1,2], Timothy W. Juliano[3], and Alex Rybchuk[2]

[1]Johns Hopkins University, Baltimore, MD, United States
[2]National Renewable Energy Laboratory, Golden, CO, United States
[3]U.S. National Science Foundation National Center for Atmospheric Research, Boulder, CO, United States

**Correspondence:** Nathan J. Agarwal (nagarw22@jh.edu)

**Abstract.** Wind resource assessments and wind power forecasts that account for wind farm wakes are sensitive to the choice of planetary boundary layer (PBL) scheme. This work compares the one-dimensional Mellor–Yamada–Nakanishi–Niino (MYNN) PBL scheme with a three-dimensional PBL (3DPBL) scheme, evaluating predictions made with both schemes against two sets of North Sea in situ observations of wind farm wakes. The optimal PBL scheme varies based on the observations (FINO1 tower

vs. aircraft), the quantity of interest (wind speed vs. turbulence kinetic energy [TKE]), and the error metric (bias, centered root mean square error [$cRMSE$], and $R^2$ vs. earth mover's distance [$EMD$]). Whereas 3DPBL wind speeds outperform MYNN wind speeds with respect to the $cRMSE$ at the FINO1 site within the turbine rotor layer, 3DPBL TKE bias underperforms MYNN TKE bias when compared to aircraft observations. Wind speeds in the aircraft region are ambiguous as to which PBL scheme is optimal. Aircraft MYNN wind speeds outperform 3DPBL wind speeds with respect to $R^2$ and $cRMSE$ but un-

derperform with respect to bias and $EMD$. Tests to determine the optimal wind farm TKE factor reveal similar variability: The aircraft observations support a wind farm TKE factor of 1 for MYNN cases and a wind farm TKE factor of 0 or 0.25 for 3DPBL cases. In contrast, the optimal wind farm TKE factor based on FINO1 observations differs by metric. For FINO1 wind speeds, the $cRMSE$ suggests that a wind farm TKE factor of 0 is most appropriate, whereas the bias and $EMD$ support a wind farm TKE factor of 1.

*Copyright statement.* This work was authored in part by the National Renewable Energy Laboratory, operated by Alliance for Sustainable Energy, LLC, for the U.S. Department of Energy (DOE) under contract no. DE-AC36-08GO28308. Funding was provided by the U.S. Department of Energy Office of Energy Efficiency and Renewable Energy Wind Energy Technologies Office. The views expressed in the article do not necessarily represent the views of the DOE or the U.S. Government. The U.S. Government retains and the publisher, by accepting the article for publication, acknowledges that the U.S. Government retains a nonexclusive, paid-up, irrevocable, worldwide license

to publish or reproduce the published form of this work, or allow others to do so, for U.S. Government purposes.





# 1 Introduction

More energy generation technologies are being developed and deployed as global energy demand continues to increase. This new energy generation is also becoming increasingly renewable. Offshore wind is one renewable technology that continues to grow, especially in the North Sea and Baltic areas (Backwell et al., 2024). Wind and wake forecasts are becoming increasingly crucial in project planning. These forecasts are sensitive to the underlying wind resource (Optis and Perr-Sauer, 2019), and meteorological wind turbine wake models are continually in development (Fischereit et al., 2022).

Wind turbine impacts on the weather are often expressed through wind farm parameterizations (WFPs) (Fischereit et al., 2022) within numerical weather prediction (NWP) models like the Weather Research and Forecasting (WRF) model (Skamarock et al., 2021). Most WFPs treat wind farms as elevated sources of turbulence and sinks of momentum (Baidya Roy et al., 2004; Fitch et al., 2012; Adams and Keith, 2013), especially after the importance of treating the wind farm as an elevated source of drag (rather than proximate to the surface) was demonstrated (Fitch et al., 2013). WFPs may include (Fitch et al., 2012) or exclude (Volker et al., 2015) an explicit source of turbulence kinetic energy (TKE), and this decision drives differences in both wind speeds and TKE (Shepherd et al., 2020; Pryor et al., 2020; Larsén and Fischereit, 2021; García-Santiago et al., 2024; Quint et al., 2024). WFPs also differ based on if and how they represent subgrid-scale processes (Abkar and Porté-Agel, 2015; Volker et al., 2015; Pan and Archer, 2018; Ma et al., 2022), which is important when multiple turbines need to be represented within one grid cell.

Wind fields predicted from NWP simulations are also sensitive to modeling choices within the WFP. Simulated wind fields depend on horizontal and vertical grid cell spacing (Lee and Lundquist, 2017; Mangara et al., 2019; Tomaszewski and Lundquist, 2020; Pryor et al., 2020), the strength of the explicit TKE source (Fitch et al., 2012; Vanderwende et al., 2016; Rajewski et al., 2016; Mangara et al., 2019; Tomaszewski and Lundquist, 2020; Siedersleben et al., 2020), the advection option (Siedersleben et al., 2020; Archer et al., 2020; Larsén and Fischereit, 2021), and the planetary boundary layer (PBL) scheme choice (Peña et al., 2023). Within the PBL scheme, parameterizations of physical quantities can further affect results. For example, the turbulence dissipation rate, $\epsilon$, affects modeled wind fields within the Mellor–Yamada–Nakanishi–Niino (MYNN) scheme (Yang et al., 2017; Bodini et al., 2020).

Given that WFPs are impacted by multiple uncertainties and these uncertainties have significant implications for power predictions, recent efforts have focused on WFP intercomparison and validation. To date, winds from WFP simulations have been validated against meteorological tower observations, aircraft observations, and lidar measurements (Draxl et al., 2014). However, WFP intercomparison and validation efforts have experienced challenges. Observations are generally staged at a distance from the wind farms that allows for validation of only the background meteorology as opposed to the wake behavior. Conclusions drawn from these validation studies may also be influenced by site-specific or meteorological conditions. Further, many previous WRF intercomparison studies contained a TKE advection bug in the Fitch et al. (2012) scheme, as identified in Archer et al. (2020).

A recent North Sea measurement campaign has stimulated interest in its potential to support WFP intercomparison and validation efforts. The Wind Park Far Field (WIPAFF) project was an aircraft expedition to understand offshore wind wake



behavior in the German Bight. This expedition took place in a location with multiple wind farms, with 41 total flights between 6 September 2016 and 15 October 2017 (Platis et al., 2018; Bärfuss et al., 2021; Larsén and Fischereit, 2021; Siedersleben et al., 2020). Siedersleben et al. (2020) leveraged 3 days of these aircraft observations that occurred during stable conditions to explore the sensitivity of grid cell spacing and TKE advection within the Fitch parameterization. Larsén and Fischereit (2021) extended the work of Siedersleben et al. (2020) by comparing the explicit wake parameterization and Fitch schemes and specifically exploring model performance during wind farm interactions with low-level jets (LLJs) and introduced wave and ocean coupling in Larsén et al. (2024). Ali et al. (2023) also used data from one day of the Siedersleben et al. (2020) case study to evaluate five (Fitch et al., 2012; Volker et al., 2015; Abkar and Porté-Agel, 2015; Pan and Archer, 2018; Redfern et al., 2019) common WFPs. Ali et al. (2023) considered the sensitivity of different parameterizations to wind speed, wind direction, TKE, wake effects, and power generation. Ali et al. (2023) then validated these parameterizations with the associated aircraft measurements as well as nearby meteorological tower and synthetic aperture radar observations.

The influence of the PBL scheme choice is one parameter that has not yet been considered for this case study and has generally been absent in the literature evaluating WFPs (Fischereit et al., 2022). Although the influence of the PBL scheme on the wind resource has been an active field of research in turbine-free NWP simulations (Zhang and Zheng, 2004; Jankov et al., 2005; Li and Pu, 2008; Nolan et al., 2009; Shin and Hong, 2011; Draxl et al., 2014), research on the impacts of the PBL scheme on turbine simulations has been limited because the default Fitch WFP has, until recently, been integrated with only the MYNN PBL scheme (Nakanishi and Niino, 2009).

However, the recent development (Kosović et al., 2020; Juliano et al., 2022; Eghdami et al., 2022) and evaluation (Arthur et al., 2022; Peña et al., 2023; Arthur et al., 2024) of the U.S. National Science Foundation National Center for Atmospheric Research (NCAR) three-dimensional PBL (3DPBL) scheme, followed by its integration with the Fitch scheme (Rybchuk et al., 2022), offers an opportunity to better understand the sensitivity of wind farm behavior to PBL scheme choice. The 3DPBL scheme is based on the Mellor and Yamada (1982) scheme and accounts for the 3D effects of turbulence by explicitly calculating the momentum, heat, and moisture flux divergences. Similar to the MYNN PBL scheme, the 3DPBL scheme is a level 2.5 model so that TKE is a prognostic variable. The 3DPBL scheme reduces errors in potential temperature, wind speed, and TKE relative to the one-dimensional (1D) MYNN scheme when compared to cold-air pool observations in the Columbia River basin (Arthur et al., 2022). The potential value of the 3DPBL scheme compared to the 1D MYNN scheme may also depend on the grid cell resolution (Peña et al., 2023).

The effects of grid cell size on turbulence representation become especially relevant in the so-called "gray zone" or "terra incognita", where the NWP horizontal grid spacing $\Delta x$ approaches a similar magnitude to the PBL depth $z$ (Wyngaard, 2004). This gray zone is relevant for wind energy applications because wind turbine applications must often consider this region where the largest turbulent eddies are neither fully parameterized (mesoscale limit; $\Delta x >> z$) nor fully resolved (large-eddy simulation limit; $\Delta x << z$) to account for both the large-scale forcings as well as the microscale turbulent eddies (Wyngaard, 2004; Chow et al., 2019; Honnert et al., 2020).

This work compares the MYNN PBL scheme with a 3DPBL scheme by validating against both tower and aircraft observations for a North Sea case study. These results address the research gap regarding the sensitivity of wake behavior to the



PBL scheme and offer guidance to the offshore forecasting community. This manuscript is organized as follows. In section 2, we describe the North Sea case study, detail the observational datasets, and outline the WRF simulation setup. In section 3, we present the results from our WRF simulations and compare these results to meteorological tower and aircraft observations associated with the case study. In section 4, we offer potential implications of the differing performance between the PBL schemes for wind resource assessments and wind power forecasting.

## 2 Methods

### 2.1 North Sea case study

The WIPAFF aircraft expedition explored the impact of several North Sea offshore wind farms (Table 1) on the atmosphere (Platis et al., 2018; Siedersleben et al., 2018a, b, 2020; Bärfuss et al., 2021) (Fig. 1). The expedition included 41 flights spanning September 2016 to October 2017, where a subset of six transect flights (Table 2) during stably stratified conditions

on 14 October 2017 has been identified as one common research case study (Siedersleben et al., 2018b, a, 2020; Larsén and Fischereit, 2021; Ali et al., 2023; Larsén et al., 2024).

**Table 1.** Select wind farm characteristics. Wind turbine performance curves are as in Ali et al. (2023). The wind farms in bold are present in immediate environs of the FINO1 and aircraft measurement regions.

| Wind farm | Hub height (m) | Diameter (m) | Turbine rating | Capacity (MW) | Number of turbines | Rated wind speed (m s$^{-1}$) |
|---|---|---|---|---|---|---|
| **Alpha Ventus ["A.V."]** | **90** | **116** | **M5000-116** | **60** | **12** | **12.5** |
| **Nordsee One** | **90** | **126** | **6.2M126** | **332** | **54** | **14** |
| **Gode** | **110** | **154** | **SWT-6.0-154** | **582** | **97** | **12** |
| Bard | 90 | 116 | M5000-116 | 400 | 80 | 12.5 |
| Global Tech | 90 | 116 | M5000-116 | 400 | 80 | 12.5 |
| **Borkum Riffgrund** | **90** | **120** | **SWT-4.0-120** | **312** | **78** | **16** |
| Meerwind | 88 | 120 | SWT-3.6-120 | 288 | 80 | 14 |
| Amrumbank West ["A.W."] | 88 | 120 | SWT-3.6-120 | 288 | 80 | 14 |
| Veja Mate ["V.M."] | 106 | 154 | SWT-6.0-154 | 402 | 67 | 12 |
| Gemini | 95 | 130 | SWT-4.0-130 | 600 | 150 | 14 |
| Riffgat | 88 | 120 | SWT-3.6-120 | 108 | 30 | 14 |
| Nordsee Ost | 95 | 126 | 6.2M126 | 295.2 | 48 | 14 |

Aircraft measurements for this 14 October 2017 case study (Bärfuss et al., 2021) were collected above the Gode wind farm (Fig. 1b and Fig. 2b) at a frequency of 100 Hz and an altitude of roughly 250 m. The flight paths across the six transects were roughly symmetrical, with transects 1, 3, and 5 traveling towards the northwest and transects 2, 4, and 6 traveling towards

the southeast (Platis et al., 2018; Bärfuss et al., 2021). Observations of wind speed, wind direction, temperature, pressure, and relative humidity were collected; here, we focus on wind speed and TKE (Table 3).

This 14 October 2017 North Sea case study also includes observations from a meteorological tower (Fig. 1b and Fig. 2a). The FINO1 tower is located immediately west of the Alpha Ventus wind farm and provides 10 min observations of wind speed, direction, pressure, and temperature. This analysis focused on wind speed and included all the available heights: 34 m, 41 m,

**Figure 1.** WRF simulation domain for the North Sea 14 October 2017 case study. (a) Three nested domains and a measurement region within the inner domain are outlined in red. (b) Measurement region with wind farms outlined in black, the FINO1 tower marked with a star, and the aircraft transect paths traced in black.





**Table 2.** Transect timings for 14 October 2017 WIPAFF case

| Transect Number | Start Time | End Time | WRF Comparison Timestep | Start Latitude/Longitude | End Latitude/Longitude |
|---|---|---|---|---|---|
| 1 | 14:20:50.860 | 14:30:12.370 | 14:30 | (53.90, 7.06) | (54.25, 6.96) |
| 2 | 14:34:41.180 | 14:44:37.520 | 14:40 | (54.25, 6.95) | (53.90, 7.06) |
| 3 | 14:48:27.970 | 14:57:43.640 | 14:50 | (53.90, 7.07) | (54.25, 6.96) |
| 4 | 15:01:38.120 | 15:11:34.970 | 15:00 | (54.25, 6.96) | (53.90, 7.06) |
| 5 | 15:45:01.130 | 15:54:05.160 | 15:50 | (53.90, 7.06) | (54.25, 6.97) |
| 6 | 15:58:29.630 | 16:08:34.810 | 16:00 | (54.25, 6.95) | (53.90, 7.06) |

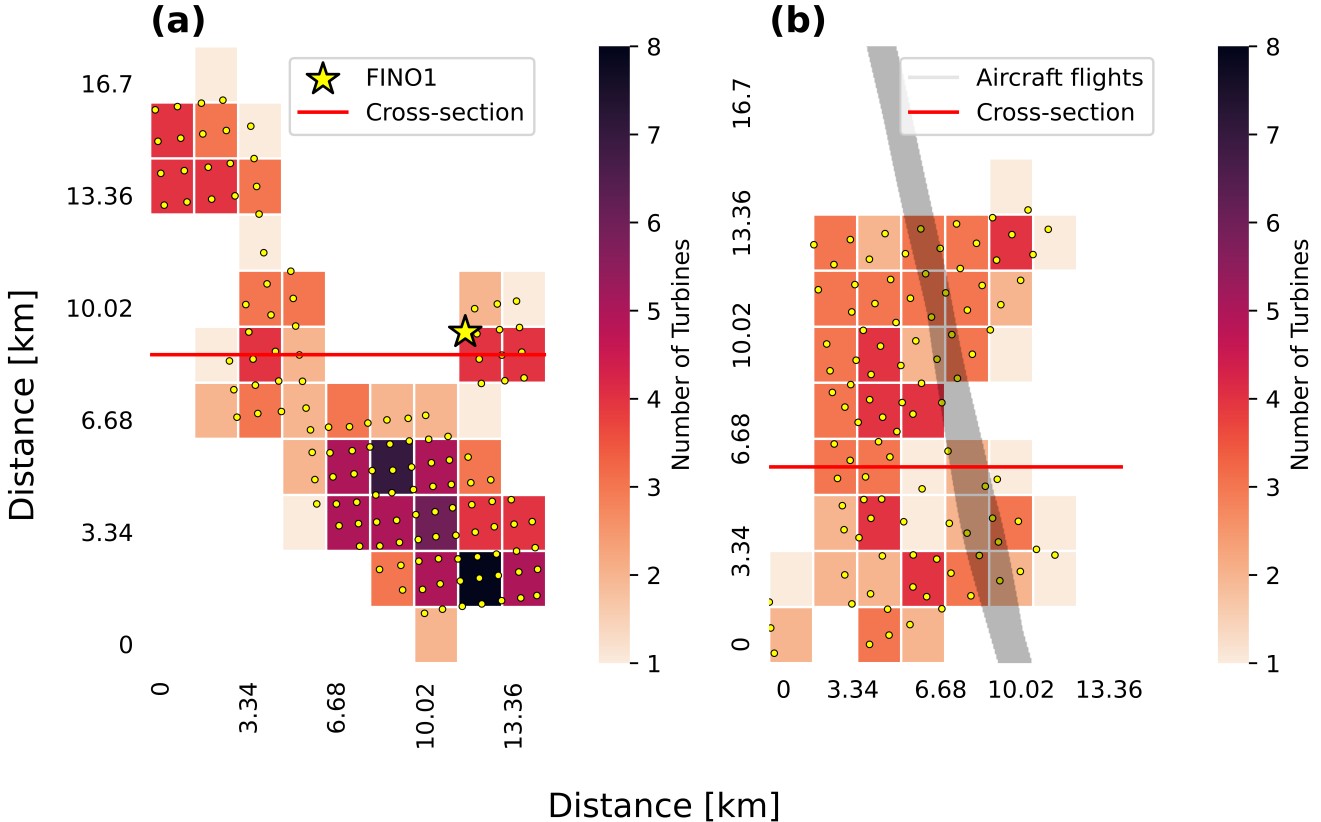

**Figure 2.** Map of the number of turbines per WRF grid cell in the innermost domain of the 1.67 km resolution for the two regions of interest within the inner region. The two axes represent the WRF grid system. (a) FINO1 site, with the tower marked with a star. (b) Aircraft site, with the six transect paths traced in black.

51 m, 61 m, 81 m, 91 m, and 102 m. TKE calculations at the FINO1 site were not available due to coarse temporal resolution of the wind observations at the FINO1 site (Table 3).



**Table 3.** In situ observations

| Site | Variable | Altitude | Temporal resolution |
|---|---|---|---|
| Aircraft | Wind Speed | 250 m | 100 Hz |
| Aircraft | TKE | 250 m | 100 Hz |
| FINO1 | Wind Speed | 34, 41, 51, 61, 81, 91, 102 m | 10 minutes |

## 2.2 Model setup

Simulations were performed with the WRF model (Skamarock et al., 2021) with the Fitch WFP (Fitch et al., 2012), modified to incorporate the 3DPBL scheme. The WRF simulations here generally followed the setup of Ali et al. (2023). Simulations represented the single day of 14 October 2017 with a 30 s timestep in the outer domain and a 10 min output, starting at 00:00:00 UTC with a 12 hour spin-up period so that the analysis period starts at 12:00:00 UTC. The region was simulated using three nested domains with an outer horizontal grid size of 15 km and a nesting ratio of 3 so that the innermost domain has a grid size of 1.67 km. Eighty vertical levels were employed to ensure sufficient vertical resolution at and below rotor height per Tomaszewski and Lundquist (2020). Specifically, 17 levels were lower than 200 m, and, depending on the turbine's diameter and hub height, between 8 and 12 levels intersected the turbine's rotor. Our model configuration also replicated many of the boundary conditions and physics options from Ali et al. (2023). The initial and boundary conditions for both analyses were represented with ERA5 reanalysis (Hersbach et al., 2020). We also followed Ali et al. (2023) by using the WRF double-moment six-class microphysics scheme (Hong et al., 2010), the RRTMG shortwave and longwave radiation scheme (Mlawer et al., 1997), and the Noah land-surface model (Niu et al., 2011) and by including the Kain–Fritsch cumulus parameterization scheme (Kain, 2004) in the outer domain only.

Differences also exist between the WRF setup presented in this work and that used in Ali et al. (2023). Here, we varied the PBL scheme to explicitly consider the influence of the PBL scheme on wake behavior for the Fitch WFP. Two PBL schemes were considered: level 2.5 MYNN ("MYNN") and the NCAR 3DPBL scheme with the PBL approximation ("3DPBL") as described in Rybchuk et al. (2022). The MYNN scheme is activated in all outer domains for all simulations. Further, Ali et al. (2023) performed their analysis on a modification of WRF v4.5.1, whereas the analysis presented in this work relied on an earlier version of WRF in which the 3DPBL scheme is integrated.

Wind farm effects were represented with the Fitch WFP. Accordingly, the drag force for a given turbine was:

$$\mathbf{F_{drag}} = \frac{1}{2}C_T\rho|\mathbf{V}|\mathbf{V}A \tag{1}$$

where $\mathbf{V}$ is the horizontal wind velocity, $C_T$ is the turbine thrust coefficient, which varies with wind speed, $\rho$ is the air density, and $A$ is the cross-sectional rotor area.

Further, the fraction of mean kinetic energy converted into TKE was governed by the turbine's thrust coefficient, $C_T$, the turbine's power coefficient, $C_P$, and a wind farm TKE factor, $\alpha$, by:

$$C_{TKE} = \alpha(C_T - C_P) \tag{2}$$



and the turbine-induced TKE tendency was:

$$\frac{\partial TKE_{ijk}}{\partial t} = \frac{0.5 N_{ij} C_{TKE} |\mathbf{V_{ijk}}|^3 A_{ijk}}{z_{k+1} - z_k} \tag{3}$$

where $i$, $j$, and $k$ are the zonal, meridional, and vertical grid cell indices, respectively; $N_{ij}$ is the turbine number density for a given cell [m$^{-2}$]; $|\mathbf{V}|_{\mathbf{ijk}}$ are the wind speed components [m s$^{-1}$]; $A$ is the turbine rotor area [m$^2$]; $C_{TKE}$ is the TKE coefficient []; and $z$ is the model level height [m].

Our simulations varied the PBL, TKE, and advection options. All simulations are summarized in Table 4. We considered TKE factors of 0, 0.25, and 1. We focused on these three wind farm TKE factors both to cover the full range of variability and to consider the 0.25 factor suggested by Archer et al. (2020).

We also considered additional model runs. Previous analyses for this work varied advection (Larsén and Fischereit, 2021), though more recent tests usually kept advection on (Sanchez Gomez et al., 2023). As such, we ran a set of model runs with the advection off. In addition, we ran "no wind farm" (NWF) simulations to distinguish turbine effects from the underlying meteorology. Results from these additional runs are presented in the Appendix. In total, we considered 16 simulations (Table 4).

The wind-speed- and turbine-model-dependent thrust and power coefficients were integrated into the WRF model through turbine-N.tbl files, where N (i.e., 1, 2, 3...) corresponds to a given turbine type (Fig. 3). Individual turbines were also integrated into the WRF grid with a windturbines.txt file from Ali et al. (2023) that contains a given turbine's latitude, longitude, and turbine type. We used files from the Ali et al. (2023) repository and extracted the key information to fit the standard Fitch WFP format.

We highlight the results from the six simulations with varying PBL scheme and wind farm TKE factor in this work (Table 4).

## 2.3 Model validation

### 2.3.1 Other diagnostic variables

We now describe other calculated quantities used to understand site performance and physical mechanisms.

The wind speed deficit, $\lambda$, characterizes wake strength:

$$\lambda_{ijk} = U_{NWF_{ijk}} - U_{Fitch_{ijk}} \tag{4}$$

where $U_{NWF_{ijk}}$ is the horizontal wind speed at a specific grid cell $ij$ and specific height index $k$ with no wind farms and $U_{Fitch_{ijk}}$ is the horizontal wind speed with the corresponding wind farm simulation.

The "drag proxy", $D$, is directly proportional to the turbine drag force (Eq. (1)) for a constant density and rotor area:

$$D = C_T V^2 \tag{5}$$

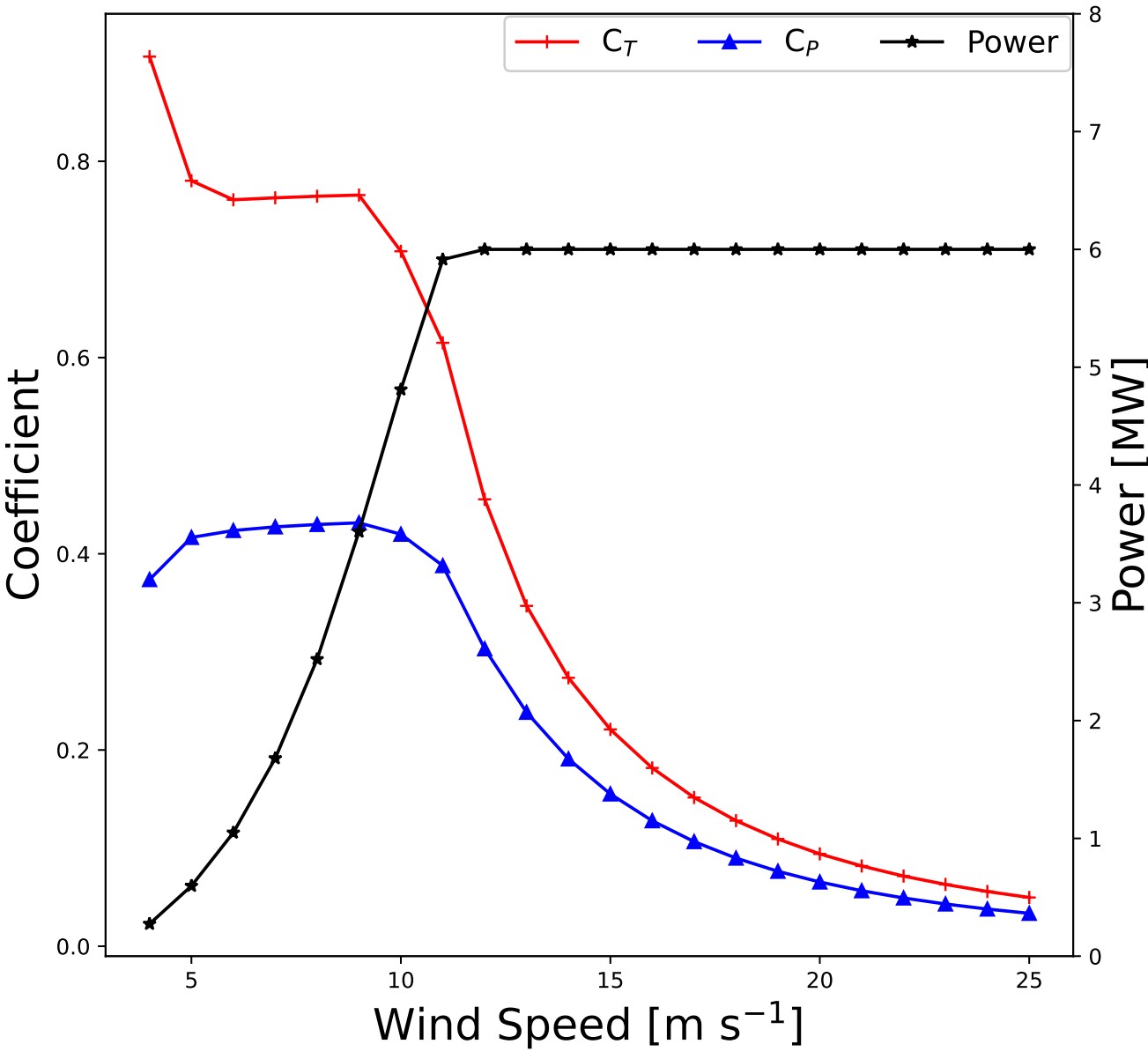

**Figure 3.** Curve illustrating turbine $C_T$, $C_P$, and power specifications for the turbine model in the Gode wind farm





**Table 4.** Full set of WRF simulations and sensitivities. The simulations in bold are those formally evaluated for performance, and the simulations not in bold are sensitivity runs explored in the Appendix.

| Simulation Name | PBL Scheme | WFP | TKE Advection | TKE Factor | Short Name |
|---|---|---|---|---|---|
| MYNN NWF Noadvect NA | MYNN | N/A | Off | N/A | mnn_NA |
| 3DPBL NWF Noadvect NA | 3DPBL | N/A | Off | N/A | 3nn_NA |
| MYNN NWF Advect NA | MYNN | N/A | On | N/A | mna_NA |
| 3DPBL NWF Advect NA | 3DPBL | N/A | On | N/A | 3na_NA |
| MYNN Fitch Noadvect 000 | MYNN | Fitch | Off | 0 | mfn_000 |
| 3DPBL Fitch Noadvect 000 | 3DPBL | Fitch | Off | 0 | 3fn_000 |
| **MYNN Fitch Advect 000** | **MYNN** | **Fitch** | **On** | **0** | **mfa_000** |
| **3DPBL Fitch Advect 000** | **3DPBL** | **Fitch** | **On** | **0** | **3fa_000** |
| MYNN Fitch Noadvect 025 | MYNN | Fitch | Off | 0.25 | mfn_025 |
| 3DPBL Fitch Noadvect 025 | 3DPBL | Fitch | Off | 0.25 | 3fn_025 |
| **MYNN Fitch Advect 025** | **MYNN** | **Fitch** | **On** | **0.25** | **mfa_025** |
| **3DPBL Fitch Advect 025** | **3DPBL** | **Fitch** | **On** | **0.25** | **3fa_025** |
| MYNN Fitch Noadvect 100 | MYNN | Fitch | Off | 1 | mfn_100 |
| 3DPBL Fitch Noadvect 100 | 3DPBL | Fitch | Off | 1 | 3fn_100 |
| **MYNN Fitch Advect 100** | **MYNN** | **Fitch** | **On** | **1** | **mfa_100** |
| **3DPBL Fitch Advect 100** | **3DPBL** | **Fitch** | **On** | **1** | **3fa_100** |

where $C_T$ and $V$ are the corresponding manufacturer-specified proper thrust coefficients and wind speeds, respectively (Fig. 3, Fig. 4).

$\Delta TKE$ represents the time-averaged difference in TKE between the two PBL schemes at a given $i$, $j$, and $k$ location.

$$\Delta TKE = TKE_{3DPBL} - TKE_{MYNN} \tag{6}$$

This difference field, consistent with all subsequent difference fields, is defined as 3DPBL - MYNN.

### 2.3.2 Spatial and temporal processing

Temporal averaging depended on the region. For the FINO1 region, 10 min WRF output data for the hours of 12:00:00–
00:00:00 UTC were averaged to compare with the FINO1 data. For the aircraft region, 10 min WRF output data for the hours of 14:10:00–16:10:00 were averaged (Table 2) to match the aircraft flights.

We also evaluated model output quantitatively, again processing each site independently and retaining the temporal resolution. For the FINO1 region, the 10 min model output data were subset to the closest $ij$ grid cell and $k$ model level to the 10 min observations. Each observational height was evaluated separately. Additional processing was necessary to capture the

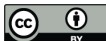

**Figure 4.** Drag proxy for each of the eight turbine models present in this case study. The solid line indicates one turbine model associated with the Gode wind farm, whereas the other drag proxy curves are dashed.





temporal and spatial variability of the aircraft transect paths. Because each transect represented approximately 10 minutes of observations, each transect could be reasonably compared to a single 10 min model output. At the same time, observations within a single transect spanned multiple model grid cells.

Given these additional considerations, we processed the aircraft region data with the following process, based on Platis et al. (2018) and Larsén and Fischereit (2021). For the wind speed comparisons, we calculated the horizontal wind speed as:

$$U = \sqrt{u^2 + v^2} \qquad (7)$$

where $u$ and $v$ are the zonal and meridional components, respectively, of the wind in m s$^{-1}$ for a given transect with the 100 Hz observations. Then, we resampled the horizontal wind speeds with a moving 2 km window. This 2 km window was first determined by Platis et al. (2018) and later implemented in Larsén and Fischereit (2021) for this case study. This window was selected based on the aircraft speed to yield an average turbulent timescale on the order of a couple of minutes. This

integral timescale appropriately separates the small-scale fluctuations from the large-scale turbulent motions (Platis et al., 2018; Larsén and Fischereit, 2021). Additional averaging was then performed, this time across grid cells. The 2 km resolution wind speed calculations were mapped to a corresponding model grid cell based on their latitude and longitude, and all the 2 km resolution wind speed calculations for a given model grid cell were averaged together. The number of 2 km resolution wind speed calculations for a given model grid cell depended on the amount of time that the aircraft spent in that grid cell.

Well-sampled grid cells may contain close to 3000 points, whereas less-sampled grid cells may contain only 10 points. These grid-cell-averaged values could then be compared to the relevant model cell value at the closest timestep (Table 2).

We employed a similar process to calculate TKE, also based on Platis et al. (2018) and Larsén and Fischereit (2021). We again isolated the 100 Hz observations for a given transect and resampled the TKE based on a 2 km moving (standard deviation) window (Platis et al., 2018):

$$TKE = \frac{1}{2}(\sigma_u^2 + \sigma_v^2 + \sigma_w^2) \qquad (8)$$

where $\sigma_u$, $\sigma_v$, and $\sigma_w$ correspond to the standard deviations of the $u$, $v$, and $w$ components, respectively. We then averaged these values across each grid cell to compare to the WRF model (Table 2).

### 2.3.3 Error metrics

The standard (Optis et al., 2020) error metrics of bias, centered root mean square error ($cRMSE$), correlation squared ($R^2$),

and earth movers' distance ($EMD$) were also calculated for each available variable at each site. The bias, $cRMSE$, and $EMD$ all have an optimal value of 0, whereas $R^2$ has an optimal value of 1. For the FINO1 region, the averages represented time averages, and for the aircraft region, the averages were across grid cells. FINO1 time averages included all 10 min data points for the 12:00:00–00:00:00 UTC period. The bias represents the difference between the modeled and observed means:

$$bias = (\overline{p} - \overline{o}) \qquad (9)$$





where $\overline{p}$ represents the modeled mean and $\overline{o}$ represents the observed mean. The $cRMSE$ represents the unbiased component of the model error. The $cRMSE$ in this case is:

$$cRMSE = [\frac{1}{N} \sum_{n=1}^{N} [(p_n - \overline{p}) - (o_n - \overline{o})]^2]^{1/2} \tag{10}$$

where $N$ is the number of data points. The Pearson correlation coefficient ($R$) represents the correspondence between two variables:

$$R = \frac{\frac{1}{N} \sum_{n=1}^{N} [(p_n - \overline{p}) - (o_n - \overline{o})]}{\sigma_p \sigma_o} \tag{11}$$

where $\sigma_p$ and $\sigma_o$ represent the standard deviations of the predictions and observations, respectively. Here, we reported the coefficient of determination, $R^2$, as recommended in Optis et al. (2020). Finally, $EMD$, also known as the Wasserstein distance, represents the area between two cumulative distribution functions and is calculated with the Python function wasserstein_distance() from the SciPy library (Virtanen et al., 2020).

### 2.3.4 Statistical significance testing

Statistical significance testing was also performed to determine the strength of the differences between simulations. This testing was performed for each error metric separately, with each transect/height representing a value of the appropriate sample. Noting the small number of data points as well as the non-normality of each sample, we prioritize our statistical testing with a Mann–Whitney U test (Mann and Whitney, 1947). We also calculate $p$ values for a traditional independent two-samples t-test (Ross and Willson, 2017) as well as for a Welch test (WELCH, 1947) for comparison in the Appendix. The main differences between these three tests are the underlying assumptions of the sample distributions. Whereas the independent two-samples t-test requires both that each sample is normally distributed and that the two samples have equal variances, the Welch test relaxes the equal variance assumption. The Mann–Whitney U test is a non-parametric, rank-sum test that further relaxes the normality requirement. In all cases, tests were performed with their corresponding Python function from the SciPy stats module (Virtanen et al., 2020), and a result is deemed statistically significant if $|p| < 0.05$.

## 3 Results

### 3.1 Site characterization

#### 3.1.1 Atmospheric stability

The modeled atmospheric stability at both the FINO1 location (Fig. 5a) and over the aircraft region (Fig. 5d) suggests a weakly stable profile near the surface with stronger stability aloft. MYNN simulations are slightly warmer than 3DPBL simulations and are slightly more stable near the surface. The wind direction profiles at both FINO1 (Fig. 5b) and the aircraft transect (Fig. 5d) regions show backing wind (i.e., the wind direction rotates from south-southwesterly near the surface to southerly aloft),



**Figure 5.** Potential temperature, wind direction, and wind speed vertical profiles from WRF simulations at a constant latitude of 54.03. In all cases, the solid black horizontal line indicates the uppermost measurement altitude, and the dashed horizontal lines indicate turbine hub height altitudes for the region. (a) FINO1 potential temperature; (b) FINO1 wind direction; (c) FINO1 horizontal wind speed; (d) aircraft potential temperature; (e) aircraft wind direction; (f) aircraft horizontal wind speed. FINO1 cases are averaged over hours 12:00:00–00:00:00, and the aircraft region cases are averaged over 14:10:00–16:10:00.





suggesting cold-air advection. Simulations with MYNN tend to have slightly more southerly winds than simulations with the 3DPBL scheme. MYNN simulations have slightly higher wind speeds at the surface, although MYNN simulations also include

lower wind speeds than 3DPBL wind speeds (Fig. 5c,f). Further, the wind speed difference between PBL schemes becomes less distinct beyond the first 200 m of the atmosphere (Fig. 5c,f). The wind direction vertical profiles for both the FINO1 (Fig. 5b) and aircraft (Fig. 5d) regions also suggest inversions at 500 m. Although these inversions are not supported by the potential temperature vertical profiles (Fig. 5a,d), these inversions are corroborated in the wind speed (Fig. 5c,f). Thus, these inversions could suggest the top of the stable boundary layer.

The stable stratification may suppress some of the turbine-generated turbulence from reaching the aircraft region measurement height, which is at least 100 m above the wind turbines (Fig. 6). Both measurement regions show TKE peaks at altitudes within the rotor region and near wind farms due to the wind-farm-generated turbulence. For the FINO1 region, the two TKE maxima align with the Riffgrund and Alpha Ventus wind farms (Fig. 6a,b). For the aircraft region, the two TKE peaks align with the Nordsee One (left in Fig. 6c,d) and Gode (right in Fig. 6c,d) wind farms. Both the FINO1 and aircraft regions also

show a greater TKE intensity with the 3DPBL scheme (Fig. 6b,d) than in the MYNN simulation (Fig. 6a,c). The stronger TKE maxima in the 3DPBL (Fig. 6b,d) at both sites also lead to greater interfarm TKE overlap than for MYNN (Fig. 6a,c), such that the TKE interactions between the wind farms are more pronounced for 3DPBL. The difference in both the intensity and degree of overlap between the 3DPBL and MYNN TKE maxima is stronger in the aircraft region (Fig. 6c,d) than in the FINO1 region (Fig. 6a,b), likely due to the larger number of wind turbines in the aircraft region. However, whereas the aircraft

region's simulation suggests a higher maximum TKE than the FINO1 region, not all of the aircraft region TKE is captured by the measurements. The turbine-induced turbulence is generated at the turbine rotor level, which is sampled well by the FINO1 tower. However, some of this turbine-induced turbulence is suppressed from reaching the aircraft region measurement height, possibly explaining the differences between simulations and observations of TKE explored below.

### 3.1.2 Spatial variability

Wind field behavior near the turbines differs from that for the rest of the simulation domain, on average. MYNN average wind speeds are higher than 3DPBL average wind speeds outside of the turbine wakes (Fig. 7a). MYNN average wind speeds likely exceed 3DPBL average wind speeds in this area because the 3DPBL scheme has higher TKE (Fig. 7b). This higher TKE with the 3DPBL scheme extracts more momentum from the mean wind, resulting in reduced wind speeds. This finding that MYNN wind speeds are higher than 3DPBL wind speeds is consistent with other comparisons of these two PBL schemes (Juliano

et al., 2022; Rybchuk et al., 2022; Arthur et al., 2022; Peña et al., 2023; Arthur et al., 2024).

In contrast, 3DPBL average wind speeds exceed MYNN Fitch average wind speeds in the turbine wakes (Fig. 7a). This distinct behavior in the wakes arises from differences in the drag forces for each PBL scheme (Fig. 4) that are very sensitive to wind speed. Because the MYNN wind speeds are slightly higher when entering the wind farms, the resulting MYNN drag force (Eq. (5)) is stronger than the 3DPBL drag force because the MYNN scheme has higher initial wind speeds and wind speeds are

below the rated wind speed. As a consequence, the MYNN scheme shows stronger and longer wakes than the 3DPBL scheme, on average (Fig. 7c). The MYNN average wind speed reduction is sufficiently strong such that 3DPBL average wind speeds



**Figure 6.** Modeled TKE cross-section at a constant latitude of 54.03. (a) FINO1 MYNN Fitch Advect 100; (b) FINO1 3DPBL Fitch Advect 100; (c) aircraft MYNN Fitch Advect 100; (d) aircraft 3DPBL Fitch Advect 100. The horizontal black line denotes the uppermost measurement height, the star indicates the FINO1 tower location, the "X" marks the first transect path, and the black circles indicate the turbine hub height.



**Figure 7.** Difference fields for inner region. (a) 3DPBL Fitch Advect 100 - MYNN Fitch Advect 100 WS; (b) 3DPBL Fitch Advect 100 - MYNN Fitch Advect 100 TKE; (c) 3DPBL Fitch Advect 100 - MYNN Fitch Advect 100 wake deficit. Turbines are marked with black circles, the FINO1 tower is marked with a yellow star, and the first transect path is marked with a solid line.

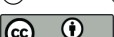



exceed MYNN average wind speeds within the turbine wake (Fig. 7a). Further, because 3DPBL average wind speeds exceed MYNN average wind speeds in this region, the 3DPBL scheme also has higher turbine-induced TKE than the MYNN scheme (Fig. 7b).

### 3.2 Measurement variability

#### 3.2.1 FINO1 tower

Wind speed differences at the FINO1 tower are consistent through the observational period. Modeled FINO1 Fitch wind speeds are generally restricted within the observational bounds and appropriately capture the temporal shifts throughout the observational period, regardless of the PBL scheme or physics options (Fig. B2). MYNN wind speeds are also consistently higher than 3DPBL wind speeds, consistent with the spatial maps (Fig. 7a). The influence of additional turbulence is also evident through the wind farm TKE factor. Increasing the wind farm TKE factor decreases the wind speeds, regardless of the PBL scheme (Fig. 8).

Observational agreement for FINO1 modeled wind speeds differs by measurement altitude. The median FINO1 modeled wind speeds at the highest locations (81 m, 91 m, and 102 m) perform best (compared to those at the lower altitudes of 34 m, 41 m, 51 m, and 61 m) for $R^2$ (Fig. 9b) and $cRMSE$ (Fig. 9c) and the worst for the bias (Fig. 9a) and $EMD$ (Fig. 9d). The low (34 m and 41 m) and middle (51 m and 61 m) heights show similar performance for the bias (Fig. 9a), $R^2$ (Fig. 9b), and $EMD$ (Fig. 9d), with some additional shaping to the $cRMSE$ (Fig. 9c). The variability in observational agreement between FINO1 heights is likely driven by wind resource variability. Whereas 3DPBL $EMD$ shifts from outperforming MYNN $EMD$ for the low and middle heights to underperforming MYNN $EMD$ for the higher heights, the physics-based trends are consistent across the FINO1 model heights, which are stronger than the variability between heights. 3DPBL cases consistently show larger wind speed biases (Fig. 9a), more variable wind speed $R^2$ (Fig. 9b), and lower wind speed $cRMSE$ (Fig. 9c) than the MYNN cases.

#### 3.2.2 Aircraft region

Similar patterns emerge for the wake-affected region of the flight path. TKE in the wake portion of the flight path is simulated to be stronger in the 3DPBL simulations than in the MYNN simulations (Fig. 10b). Further, in this middle section of the flight path, increasing the wind farm TKE factor increases the TKE, regardless of the PBL scheme (Fig. 10b). These two trends are likewise mirrored in the wind speed patterns in the middle of the flight path. Because the 3DPBL scheme has higher TKE, the 3DPBL scheme consequently shows lower wind speeds (Fig. 7b, 10a). Further, increasing the wind farm TKE factor further reduces the wind speeds during this portion of the path, regardless of the PBL scheme (Fig. 10a). Whereas increasing the wind farm TKE factor is the dominant influence for TKE performance, wind speed is affected by multiple meteorological features in nonlinear ways. As such, the wind speed flight path patterns exhibit greater variability (Fig. 10a).

Simulated aircraft region wind speeds and TKE are also likely subject to systematic influences between transects. First, a performance discrepancy between odd and even transects exists. Modeled wind speeds and TKE for southeast-to-northwest



**Figure 8.** Time series of modeled horizontal wind speeds (WS) compared to 81 m FINO1 observations for the hours of 12:00:00–00:00:00. Both the modeled wind speeds and observed wind speeds are resampled to 30 minutes. The analogous time series for other altitudes are provided in the Appendix.



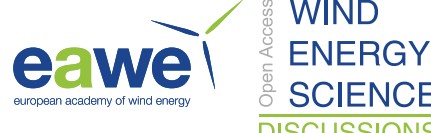

**Figure 9.** Error metric box plot for FINO1 region wind speeds across all Fitch (advection) simulations. These simulations include the wind farm TKE factors 0, 0.25, and 1. The box encloses the interquartile range (IQR), and the whiskers extend to Q1-1.5*IQR and Q3+1.5*IQR. (a) Bias; (b) $R^2$; (c) $cRMSE$; (d) $EMD$.



**Figure 10.** Aircraft region (250 m) simulated and observed (a) wind speed and (b) TKE across the flight path for transect 1.



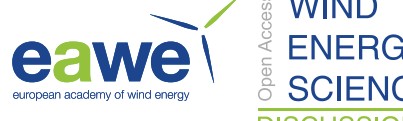

**Figure 11.** Wind speed error metrics by transect across all Fitch (advection) simulations. These simulations include the wind farm TKE factors 0, 0.25, and 1. The box encloses the interquartile range (IQR), and the whiskers extend to Q1-1.5*IQR and Q3+1.5*IQR. (a) Bias; (b) $R^2$; (c) $cRMSE$; (d) $EMD$.





**Figure 12.** TKE error metrics by transect across all Fitch (advection) simulations. These simulations include the wind farm TKE factors 0, 0.25, and 1. The box encloses the interquartile range (IQR), and the whiskers extend to Q1-1.5*IQR and Q3+1.5*IQR. (a) Bias; (b) $R^2$; (c) $cRMSE$; (d) $EMD$.





transects (1, 3, and 5) have a larger bias than modeled wind speeds and TKE for northwest-to-southeast transects (2, 4, and 6)
with respect to the bias (Fig. 11a, Fig. 12a). This systematic transect variability corresponds to the reversed directions of the
transect paths – whereas transects 1, 3, and 5 are performed in the northwesterly direction, transects 2, 4, and 6 are performed
in the southeasterly direction. This directional variability likely corresponds to differences in relative sensor alignment.

# 4 Model evaluation

## 4.1 Effect of PBL scheme

**Table 5.** $p$ values according to the Mann–Whitney U test. A bold cell indicates statistical significance at $|p| <= 0.05$.

| Parameter | Quantity | Bias | $R^2$ | $EMD$ | $cRMSE$ |
|---|---|---|---|---|---|
| PBL | FINO wind speed | 0.4065 | 0.3391 | 0.5629 | **0.0** |
| PBL | Aircraft region wind speed | **0.0119** | **0.0** | **0.0184** | **0.0** |
| PBL | Aircraft region TKE | **0.0** | 0.937 | 0.1032 | 0.3843 |
| $C_{TKE}$ | FINO wind speed | **0.0** | **0.0013** | **0.0007** | **0.003** |
| $C_{TKE}$ | Aircraft region wind speed | 0.1124 | 0.8399 | 0.2855 | 0.8852 |
| $C_{TKE}$ | Aircraft region TKE | **0.0304** | **0.0194** | 0.4025 | **0.0017** |

A primary goal of this effort is to explore how simulated wake behavior changes based on the PBL scheme. The optimal PBL
scheme depends on the site, error metric, and variable considered. The statistical significance of these differences, however, is
limited. A statistically significant difference between 3DPBL and MYNN exists for wind speeds at the FINO1 site according
to the $cRMSE$ (Table 5) but not for the bias, $R^2$, or $EMD$. TKE in the aircraft region also shows a statistically significant
difference for a single metric – in this case, the bias (Table 5). In contrast, a statistically significant difference between 3DPBL
and MYNN is present for all four metrics within the aircraft region (Table 5).

In the FINO1 region, 3DPBL wind speeds outperform MYNN wind speeds in representing wind speeds with respect to
$cRMSE$ (Fig. 13b), while the bias (Fig. 13a), $R^2$ (Fig. 13c), and $EMD$ (Fig. 13d) are comparable between the PBL schemes.
In contrast, the MYNN scheme holds a more decisive lead in comparison to observations collected 250 m above the surface
and 100 m above a wind farm. MYNN TKE outperforms 3DPBL TKE with respect to bias (Fig. 15a) and $EMD$ (Fig. 15d),
with no clear winner for $cRMSE$ (Fig. 15b) or $R^2$ (Fig. 15c). For wind speeds evaluated with the aircraft dataset, MYNN
outperforms 3DPBL with respect to $cRMSE$ (Fig. 14b) and $R^2$ (Fig. 14c), whereas 3DPBL outperforms MYNN with respect
to bias (Fig. 14a) and $EMD$ (Fig. 14d). The swapped error metric performance between wind speed and TKE in the aircraft
region may reflect competing optimizations. Adding additional TKE induces more mixing, extracts more momentum, and
slows the winds. This mechanism allows high wind farm TKE factors to improve the wind speed bias in the aircraft region.
However, the additional TKE is not necessarily physical. Whereas the MYNN cases with a wind farm TKE factor of 1 improve





both the wind speed bias and the TKE bias, the 3DPBL cases with a wind farm TKE factor of 1 improve the wind speed bias at the expense of the TKE bias.

Previous studies have also found that the optimal PBL scheme depends on the model, site, and metric constraints. Draxl et al. (2014) found that the optimal PBL scheme depended on the atmospheric stability in an analysis of the Høvsøre wind farm.
Hahmann et al. (2020) found that the MYNN scheme outperformed the Yonsei University (YSU) scheme in weekly simulations that used spectral nudging in characterizing eight sites in Northern Europe. Hahmann et al. (2020) also performed a series of 25 simulations to compare PBL schemes, land surface models, and surface layer options and found that the differences between the sites were larger than the differences between PBL schemes, regardless of the error metric. Sheridan et al. (2024) found that the MYNN-based dataset both identified more of the extremely LLJ events and had a higher rate of false LLJ identification
than the YSU-based dataset in the North Atlantic. Storm et al. (2009) similarly forecast LLJs in the Great Plains and found that, although the YSU scheme outperformed the Mellor–Yamada–Janjic (MYJ) PBL scheme in forecasting the wind direction, the MYJ scheme outperformed the YSU scheme with respect to wind speed for the same West Texas case. Storm et al. (2009) also ascribe these differences to be largely site-dependent and argue that the results would likely be different at another site. Smith et al. (2018) also analyzed LLJs and found the vertical grid cell spacing to affect model performance, especially for the MYNN
scheme. Similarly, Peña et al. (2023) found that the optimal PBL scheme depended on the horizontal grid cell spacing.

Our results, highlighting the advantages of the 3DPBL scheme within the turbine rotor layer, are also supported by previous MYNN and 3DPBL intercomparisons. Arthur et al. (2022) found that the 3DPBL scheme consistently outperformed the MYNN scheme for a Columbia River Gorge site. Similarly, Arthur et al. (2024) compared the MYNN and 3DPBL schemes for an Altamont Pass site and also found the 3DPBL scheme to reduce wind speed error. Our results in the turbine rotor layer
align with those of Arthur et al. (2022) in that they both support the 3DPBL scheme over the MYNN scheme in stably stratified conditions. Our results also align with those of Arthur et al. (2024) in that they both show the 3DPBL scheme outperforming the MYNN scheme within the turbine rotor layer. Notably, whereas Arthur et al. (2022) and Arthur et al. (2024) were both performed in areas of complex terrain, which has been theorized to provide a theoretical advantage to the 3DPBL scheme (Kosović et al., 2020; Juliano et al., 2022), this work considers the relatively homogeneous offshore area, suggesting that the
advantages of the 3DPBL scheme are not confined to complex terrain.

Differences in the relative measurement height between the two sites in this case study may also affect the PBL comparison. Whereas the FINO1 tower is within the turbine rotor region, the aircraft measurements are taken more than 100 m above the turbines in a stably stratified boundary layer (Fig. 5) that suppresses some interactions between the atmosphere sampled at the turbine level and the aircraft level (Fig. 6). Thus, the 3DPBL scheme improves turbine-induced turbulence characterization in
the turbine rotor region as sampled at FINO1 (Fig. 14a) and overpredicts turbine-induced turbulence aloft as sampled by the aircraft measurements (Fig. 15a). Other site-based considerations are less likely to be driving the relative PBL performance trends. The documented even/odd transect variability (Fig. 11, Fig. 12) is not likely to drive the PBL-based differences given that the PBL-based differences are larger than the transect variability. In addition, although the FINO1 region includes 12 hours of observations and the aircraft region includes only roughly 2 hours of observations, 3DPBL is still optimal over MYNN if we





**Figure 13.** Wind speed error metric box plot for the FINO1 tower measurements. The box and whiskers describe FINO1 model height variability and are based on Q1 (25th percentile), Q3 (75th percentile), and the interquartile range (IQR) (Q3–Q1). The box encloses the IQR, and the whiskers extend to Q1-1.5*IQR and Q3+1.5*IQR. The simulation names are mapped according to the short names provided in Table 4, and the vertical dotted lines visually separate simulations by wind farm TKE factor. (a) Wind speed bias; (b) wind speed $cRMSE$; (c) wind speed $R^2$; (d) wind speed $EMD$.


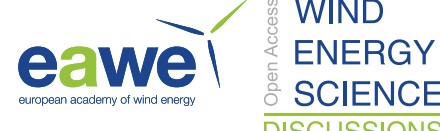

**Figure 14.** Error metric box plot for aircraft observations collected at 250 m. The box and whiskers describe aircraft transect variability and are based on Q1 (25th percentile), Q3 (75th percentile), and the interquartile range (IQR) (Q3–Q1). The box encloses the IQR, and the whiskers extend to Q1-1.5*IQR and Q3+1.5*IQR. The simulation names are mapped according to the short names provided in Table 4, and the vertical dotted lines visually separate simulations by wind farm TKE factor. (a) Wind speed bias; (b) wind speed $cRMSE$; (c) wind speed $R^2$; (d) wind speed $EMD$.



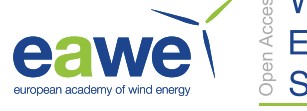

**Figure 15.** Error metric box plot for aircraft observations collected at 250 m. The box and whiskers describe aircraft transect variability and are based on Q1 (25th percentile), Q3 (75th percentile), and the interquartile range (IQR) (Q3–Q1). The box encloses the IQR, and the whiskers extend to Q1-1.5*IQR and Q3+1.5*IQR. The simulation names are mapped according to the short names provided in Table 4, and the vertical dotted lines visually separate simulations by wind farm TKE factor. (a) TKE bias; (b) TKE $cRMSE$; (c) TKE $R^2$; (d) TKE $EMD$.





reduce our FINO1 evaluation timeframe to match that of the aircraft region. Finally, the PBL-based preferences are reinforced at all heights of the FINO1 tower (Fig. B2a,c,e), even despite performance variability between FINO1 heights (Fig. 9).

## 4.2  Effect of wind farm TKE factor, $C_{TKE}$

Recent scientific discussion has focused on the determination of the optimal value of the wind farm TKE factor, $C_{TKE}$, with varying conclusions, and the observations and simulations presented here do not resolve this debate. The optimal wind farm
TKE factor, $C_{TKE}$, in Eq. (2) depends on the PBL scheme, the site, and the metric, but only some of these differences are statistically significant. Wind speeds in the FINO1 region differ with statistical significance (Table 5) according to all four metrics. Further, TKE in the aircraft region shows statistically significant differences between the minimal and maximal wind farm TKE factors for the bias, $R^2$, and $cRMSE$ (Table 5). In contrast, wind speeds in the aircraft region (Table 5) show statistically insignificant differences between the minimal and maximal wind farm TKE factors, regardless of the metric.
Although the wind speed results in the region above the wind farm are statistically insignificant, they do align with those for TKE. For the aircraft observations above the turbines, a wind farm TKE factor of 1 is most appropriate for both MYNN wind speeds and TKE, whereas the 3DPBL wind speeds and TKE benefit from 0–0.25 wind farm TKE factors (Fig. 14a,b,d, Fig. 15a,b,d).

$C_{TKE}$ discrepancies with PBL preference in the aircraft region may best be explained by how the two PBL schemes char-
acterize turbulence. The 3DPBL scheme consistently has larger TKE than the MYNN scheme (Fig. 15a). This discrepancy in baseline turbulence levels affects how the two PBL schemes respond to an increasing $C_{TKE}$. The MYNN scheme moves from underpredicting TKE to more appropriately predicting TKE as $C_{TKE}$ increases (Fig. 15a). In contrast, the 3DPBL scheme overpredicts the TKE and exacerbates the TKE overprediction with larger $C_{TKE}$ (Fig. 15a). This trend is mirrored with aircraft region wind speed bias (Fig. 14a).
The appropriate wind farm TKE factor for comparison to the FINO1 observations differs between metrics. Both the wind speed bias (Fig. 13a) and wind speed $EMD$ (Fig. 13d) support a $C_{TKE}$ of 1, whereas the $cRMSE$ (Fig. 13b) suggests that a $C_{TKE}$ of 0 is more appropriate. $R^2$ (Fig. 13c) shows no clear optimal $C_{TKE}$. This metric distinction holds for both PBL schemes (Fig. 13). The one caveat in the FINO1 region is that, with increasing $C_{TKE}$, wind speeds increase, and both PBL schemes arrive at an appropriate (according to the bias) wind speed characterization (Fig. 13a). In the FINO1 domain, increased
turbulence in the rotor region extracts more momentum from aloft into the measurement region and increases wind speeds. The split metric performance signals that improving the mean wind speed does not necessarily improve wind speed predictions at individual locations.

Our performance differences based on $C_{TKE}$ are consistent with others reported in the literature. Several works corroborate improved performance with increased $C_{TKE}$. Vanderwende et al. (2016) found that Fitch simulations with a wind farm TKE
factor of 0 underpredicted TKE. García-Santiago et al. (2024) similarly showed that the Fitch scheme could improve TKE underpredictions. Siedersleben et al. (2020), Ali et al. (2023), and Sanchez Gomez et al. (2023) also identified improved predictions with larger $C_{TKE}$. However, the overestimation of $C_{TKE}$ is also evident in the literature. Vanderwende et al. (2016) likewise found that a wind farm TKE factor of 1 contributed to TKE overpredictions. Similarly, García-Santiago et al.





(2024) found that the Fitch scheme with a wind farm TKE factor of 1 overpredicted TKE. Mangara et al. (2019) and Larsén
et al. (2024) likewise identified excess TKE predictions with a wind farm TKE factor of 1. One challenge in comparing our
results with those in Vanderwende et al. (2016), Mangara et al. (2019), and Siedersleben et al. (2020) is that these works
were completed prior to the advection bug identification. However, the results presented in Siedersleben et al. (2020) are also
corroborated in the post-bug-identification work of Larsén and Fischereit (2021).

Our results also do not support the Archer et al. (2020) recommendation of a universal wind farm TKE factor of 0.25. With
the same (MYNN) PBL scheme as Archer et al. (2020), we find that the optimal wind farm TKE factor of 1 is more appropriate
to characterize the aircraft region. The differences in the optimal wind farm TKE factor between these two studies may arise
from stability differences between the sites. Whereas the present study focuses on stably stratified conditions, Archer et al.
(2020) was performed under neutral conditions. This contradiction is further corroborated by Larsén and Fischereit (2021)
and Ali et al. (2023), who both found that the 0.25 wind farm TKE factor is non-optimal for this case study. Our results also
demonstrate the variability of the optimal wind farm TKE factor. We show that although the optimal wind farm TKE factor
for 3DPBL cases in the aircraft region is 0.25, the optimal wind farm TKE factor in the aircraft region for MYNN cases is 1.
We also show that the optimal wind farm TKE factor varies by site and metric. At the FINO1 site, the optimal wind farm TKE
factor is either 0 or 1 depending on the error metric. This variability is also corroborated by Ali et al. (2023),Optis (2024), and
Larsén et al. (2024) who also emphasize the variability of an optimal wind farm TKE factor.

## 5  Conclusions

This work addresses model capabilities for representing wind farm wakes with the Fitch WFP in the WRF model. This question
underpins an important and understudied sensitivity of the NWP models that support both wind power forecasting and wind
energy assessment. This work explores this question as one of the first comparative evaluations between the 1D MYNN PBL
scheme and the newly-Fitch-integrated NCAR 3DPBL scheme against two sets of in situ observations for an offshore case
study.

The optimal PBL scheme depends on the site. For wind speeds modeled at the site within the turbine rotor region, 3DPBL
outperforms MYNN with respect to the $cRMSE$. In contrast, for TKE modeled within a region 100 m above a wind farm,
MYNN outperforms 3DPBL with respect to the bias. Ambiguously, for wind speeds modeled within the region 100 m above
a wind farm, MYNN wind speeds outperform 3DPBL wind speeds with respect to $R^2$ and $cRMSE$ but underperform with
respect to bias and $EMD$. These site-based differences in the optimal PBL scheme likely reflect differences in the relative
measurement height between the two sites. Whereas the FINO1 observations are collected within the turbine rotor region,
the aircraft observations are collected 100 m above the turbine rotor layer, such that interactions between the turbine-induced
turbulence and the aircraft measurements may be constrained by a stable boundary layer. Thus, 3DPBL simultaneously more
appropriately characterizes behavior within the turbine rotor layer and overpredicts the amount of TKE reaching the aircraft
region site.

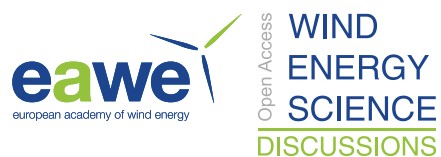

The optimal wind farm TKE factor, $C_{TKE}$, is similarly nonuniversal. For wind speeds modeled at the site within the turbine rotor region, the bias and $EMD$ support a $C_{TKE}$ of 1, whereas the $cRMSE$ suggests a $C_{TKE}$ of 0. In contrast, for both wind speeds and TKE modeled at the site 100 m above a wind farm, the optimal $C_{TKE}$ depends on the PBL scheme. MYNN cases above the Gode wind farm find the best agreement with observations with a $C_{TKE}$ value of 1, whereas 3DPBL cases above

that wind farm support a $C_{TKE}$ of 0 or 0.25. The diverging optimizations by PBL scheme reflect differences in the baseline turbulence levels for the two PBL schemes. 3DPBL consistently shows larger TKE than MYNN. As such, injecting more TKE with MYNN always improves performance, and TKE injections with 3DPBL overpredict TKE.

Subsequent investigations could explore other case studies to provide perspective into the generalizability of the results across other sites. Similarly, datasets from the third Wind Forecast Improvement Project (WFIP3) could be useful to explore

how offshore wind characterization might differ between the North Sea and the eastern United States (WFIP3). Moreover, datasets from the land-based, horizontally homogeneous American WAKE experimeNt – or AWAKEN – campaign (Moriarty et al., 2024) could be useful to study because previous land-based studies analyzing the 3DPBL scheme have involved complex terrain.

Further, this study focuses on simulations of the atmosphere alone, without coupling to the ocean surface or the water

below. Another example of future work may include performing this analysis with an ocean model coupling (Raghukumar et al., 2022, 2023; Daewel et al., 2022). Although introducing a coupled ocean–atmosphere model does not always improve performance (Gaudet et al., 2024), introducing ocean coupling to this model could help diagnose sources of wind speed error for this model. Introducing a coupled ocean–atmosphere model also provides physical insight by potentially affecting the relative performance between the PBL schemes. Introducing ocean coupling also provides a more direct comparison to the

work of Larsén et al. (2024), which included atmosphere–wave coupling, for this case study. Recreating this analysis with an additional ocean model like the Regional Ocean Modeling System could provide insight into the influence of oceanic forcings on wind wake behavior and the resulting consequences on surface currents.

*Data availability.* The FINO1 data can be downloaded here: https://insitu.bsh.de/rave/index.jsf?content=insitu, and the WIPAFF aircraft data can be downloaded here: https://doi.pangaea.de/10.1594/PANGAEA.903088. Post-processing code is available here: 10.5281/zen-
odo.14751600. The WRF code used in this analysis is based on the publicly-available version of 3DPBL code provided here: <update on acceptance>

## Appendix A:  Additional model runs

### A1  Contrast between NWF and WF options

Average wind speeds for the Fitch and NWF cases are similar in regions without wind turbines and differ near the turbines.
Both NWF (Fig. A1a,c) and Fitch (Fig. A1b,d) average wind speeds are lower over land than over water. Further, the fastest average wind speeds are immediately to the west of the coast for both the NWF (Fig. A1a,c) and Fitch (Fig. A1b,d) cases.





**Figure A1.** Horizontal wind speeds for the inner 1.67 km domain. (a) 3DPBL NWF Advect NA. (b) 3DPBL Fitch Advect 100. (c) MYNN NWF Advect NA. (d) MYNN Fitch Advect 100. (e) 3DPBL NWF Advect NA - MYNN NWF Advect NA. (f) 3DPBL Fitch Advect 100 - MYNN Fitch Advect 100. Turbines are marked with black circles, the FINO1 tower is marked with a yellow star, and the first transect path is marked with a solid line.





**Table A1.** $p$ values according to the Mann–Whitney U test. A bold cell indicates statistical significance at $|p| <= 0.05$.

| Parameter | Quantity | Bias | $R^2$ | $EMD$ | $cRMSE$ |
|-----------|----------|------|-------|-------|---------|
| WF Option | FINO wind speed | 0.0 | **0.02776** | 0.0 | 0.0559 |
| WF Option | Aircraft region wind speed | **0.0** | **0.0** | **0.0** | **0.0** |
| WF Option | Aircraft region TKE | **0.0** | **0.0** | **0.0** | **0.0** |

The predominant westerly wind direction and the significant distance of the measurement regions from the coast demonstrate that the measurement regions are not influenced by the coastal decelerations. MYNN average wind speeds (Fig. A1c,d) are also consistently higher than 3DPBL average wind speeds (Fig. A1a,b) for both NWF and Fitch cases outside of the turbine wakes (Fig. A1e,f). This difference in wind speeds between the PBL schemes can be explained by TKE differences between the two PBL schemes. Because the 3DPBL scheme has larger TKE (Fig. 7b), the 3DPBL scheme extracts more momentum and reduces wind speeds. In contrast, Fitch and NWF wind speeds differ near the turbines. Notably, 3DPBL Fitch average wind speeds exceed MYNN Fitch average wind speeds in the turbine wakes (Fig. A1f). This reversal of which PBL scheme shows the higher average wind speed can be explained by differences in the turbine drag force between the two PBL schemes. The MYNN scheme has a stronger turbine drag force (Eq. (5), Fig. 4) because of its higher initial wind speeds, which also implies that the MYNN scheme has stronger and deeper wakes (Fig. 7c). This reversal of which PBL scheme shows the higher average wind speed is isolated to the monotonically increasing region of the drag (proxy) curve (Fig. 4). If the wind speeds were instead within the monotonically decreasing region of the drag (proxy) curve (Fig. 4), MYNN wind speeds would likely exceed 3DPBL wind speeds even in the wakes. Given that NWF wind speeds mirror Fitch wind speeds outside of the turbine wakes and NWF wind speeds differ from Fitch wind speeds within the wakes, the dominant mechanism for these differences is more likely related to the turbines and not to the underlying meteorology.

## A2  Effect of TKE advection

**Table A2.** $p$ values according to the Mann–Whitney U test. A bold cell indicates statistical significance at $|p| <= 0.05$.

| Parameter | Quantity | Bias | $R^2$ | $EMD$ | $cRMSE$ |
|-----------|----------|------|-------|-------|---------|
| Advection | FINO wind speed | 0.9467 | **0.0009** | 0.7248 | **0.0005** |
| Advection | Aircraft region wind speed | 0.9971 | 0.5553 | 0.8289 | 0.6157 |
| Advection | Aircraft region TKE | 0.3892 | 0.0508 | 0.8061 | 0.1340 |

The appropriate treatment of the TKE advection option for simulations of wind farm impacts – and specifically this case study – has received much discussion in the literature. Most recently, Ali et al. (2023) argue that the advection option should be turned off. Ali et al. (2023) make this judgment based on the Siedersleben et al. (2020) finding that performance in the aircraft region improved by turning the advection option off. However, Siedersleben et al. (2020) performed their analysis with a version of the Fitch scheme that included the advection bug. As such, the results presented in Siedersleben et al. (2020) are





qualitatively different from those presented in Larsén and Fischereit (2021). Although Larsén and Fischereit (2021) performed their analysis after the advection bug was addressed and performed simulations with advection both on and off, Larsén and

Fischereit (2021) argue that further analysis would be necessary to make a formal recommendation.

Our results reveal performance differences based on the advection option for the FINO1 site but not for the aircraft region site. Performances for both wind speed and TKE in the aircraft region are largely unaffected by the advection option, regardless of whether we consider the bias (Fig. B11a, Fig. B12a), $cRMSE$ (Fig. B11b, Fig. B12b), $R^2$ (Fig. B11c, Fig. B12c), or $EMD$ (Fig. B11d, Fig. B12d). In contrast, the FINO1 site has a more nuanced response to the advection option. Neither the FINO1

bias (Fig. B13a) nor the $EMD$ (Fig. B13d) are affected by the advection option. However, the FINO1 $cRMSE$ (Fig. B13b) and $R^2$ (Fig. B13c) show advection-based responses that are mediated by the PBL scheme and the wind farm TKE factor. MYNN wind speeds decrease by introducing advection according to the $cRMSE$ (Fig. B13b) and $R^2$ (Fig. B13c). In contrast, 3DPBL wind speeds improve with advection according to the $cRMSE$ (Fig. B13b) and $R^2$ (Fig. B13c) for $C_{TKE}$ of 0 and 0.25 and worsen the $cRMSE$ (Fig. B13b) and $R^2$ (Fig. B13c) for a $C_{TKE}$ of 1.

The site-based responses to the advection option may reflect locational differences. The results at the FINO1 location show greater sensitivity to the advection option than those at the aircraft measurement site. First, the FINO1 site is marked by a single grid cell at a wind farm's edge (Fig. 2a) and surrounded by other cells with wind turbines. Slight horizontal adjustment of this FINO1 tower location would capture the local TKE source as opposed to the sink (Fig. A2c,d). In addition, because the FINO1 performance is defined by a single grid cell, local TKE imbalances between grid cells are not compensated in

averaging calculations. In contrast, the aircraft measurements experience several competing interfarm advection pools that are compensated across a broader domain (Fig. A2a,b). The two sites are also at different measurement heights. As explored earlier, the FINO1 site more directly interacts with the turbine-induced turbulence at the rotor level than does the aircraft region site (Fig. 6). As such, the FINO1 site is more sensitive to TKE differences. Although introducing advection does support some vertical transport of TKE at the aircraft measurement locations (Fig. A3a,b), maximum TKE is still better physically aligned

with the measurement altitude for the FINO1 observations (Fig. A3c,d). As a consequence, the FINO1 region measurements show greater sensitivity to the advection option than the aircraft region measurements.

Differences in the amount of TKE generated by the PBL scheme may explain PBL-based differences in FINO1 advection performance. Local TKE imbalances are stronger for 3DPBL TKE (Fig. A2a,c, Fig. A3a,c) than for MYNN TKE (Fig. A3b,d, Fig. A3b,d). Because the 3DPBL scheme has larger TKE (Fig. 7b), the 3DPBL scheme is more sensitive to TKE movement

throughout the region. This higher baseline TKE also explains why the anomalous performance degradation from the advection option is restricted to the 100% $C_{TKE}$ simulation for 3DPBL cases (Fig. 15a). In this situation, TKE is already overpredicted, and introducing advection further pushes even more TKE into the measurement region.

We recommend using TKE advection for this case study, in contrast to prior recommendations. Our results do not align with those of Siedersleben et al. (2020) or Ali et al. (2023). However, as Larsén and Fischereit (2021) also note, pre-bug and

post-bug advection patterns are significantly different, and rigorous comparison of the two is speculative. In contrast, Larsén and Fischereit (2021) completed their analysis after the advection bug identification. Our results show a mixed agreement with those of Larsén and Fischereit (2021). Our results agree with those in Larsén and Fischereit (2021) in not identifying significant



**Figure A2.** Horizontal slices to mark TKE differences between cases with advection on and cases with advection off. Red indicates that TKE is higher without advection. (a) Aircraft region 3DPBL Fitch 100. (b) Aircraft region MYNN Fitch 100. (c) FINO1 3DPBL Fitch 100. (d) FINO1 MYNN Fitch Advect 100. FINO1 cases are averaged over hours 12:00:00–00:00:00, and the aircraft region cases are averaged over 14:10:00–16:10:00. Turbines are marked with black circles, the FINO1 tower is marked with a yellow star, and the first transect path is marked with a solid line.

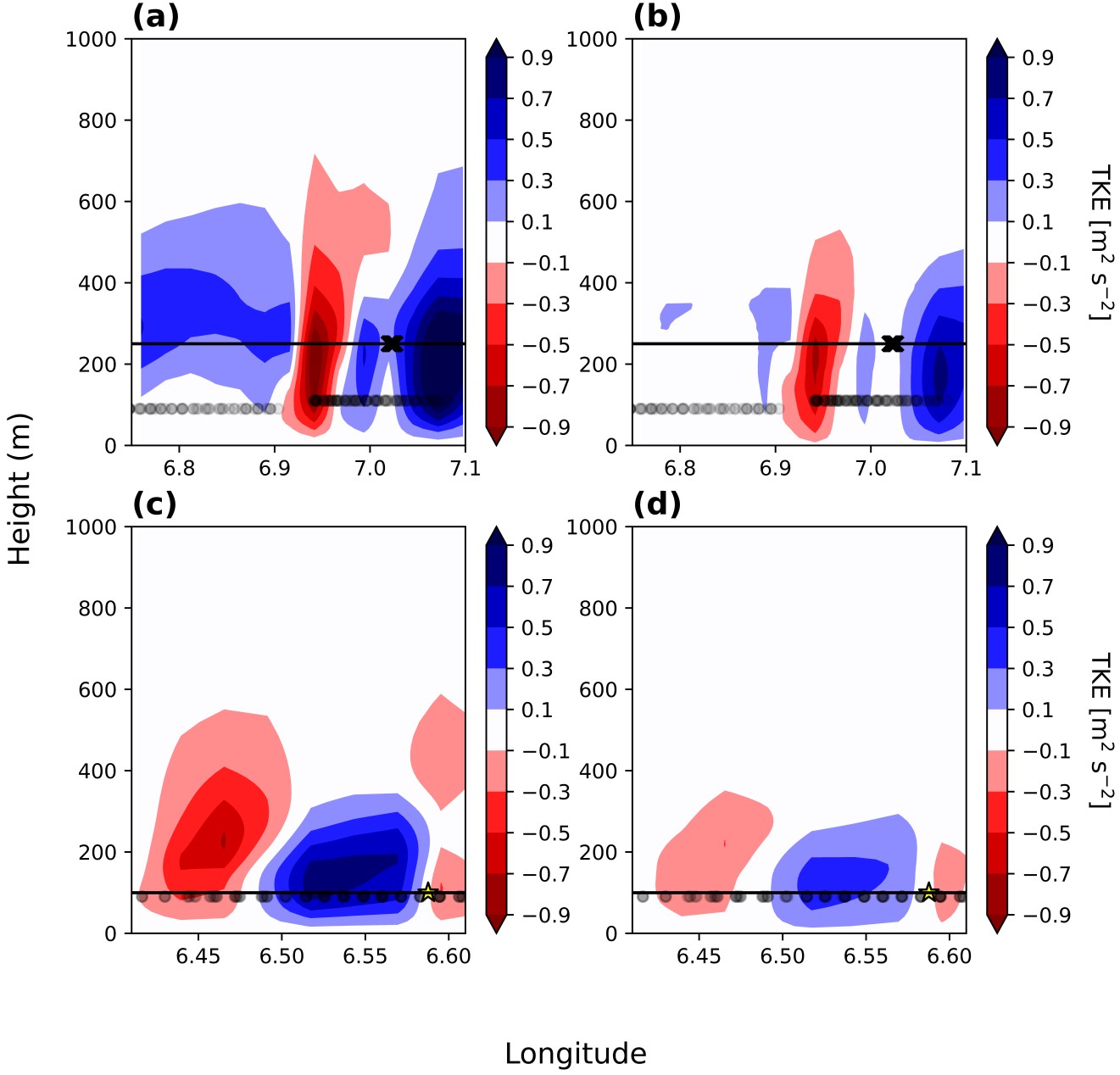

**Figure A3.** Vertical slices at a constant latitude of 54.03 to mark TKE differences between cases with advection on and cases with advection off. Red indicates that TKE is higher without advection. (a) Aircraft region 3DPBL Fitch 100. (b) Aircraft region MYNN Fitch 100. (c) FINO1 3DPBL Fitch 100. (d) FINO1 MYNN Fitch Advect 100. FINO1 cases are averaged over hours 12:00:00–00:00:00, and the aircraft region cases are averaged over 14:10:00–16:10:00. Turbines are marked with black circles, the FINO1 tower is marked with a yellow star, and the transect path is marked with an "X".




performance differences in wind speed or TKE based on the advection option in the aircraft region. However, we do observe performance improvements by introducing advection in the FINO1 region. Further, our results agree with those in Larsén and Fischereit (2021) in identifying significant TKE differences between the cases corresponding to upwind and downwind of the Gode wind farm. However, Larsén and Fischereit (2021) argue that further analysis would be necessary to properly assess whether the advection option could appropriately characterize this site. In contrast, we argue that the advection option should be included on two bases. First, we note that TKE advection was omitted only from early versions of MYNN for reasons of numerical stability, not on a physical basis (Olson et al., 2019). Second, we note the guidance from Wadler et al. (2023), who show that introducing TKE advection allows for a more realistic distribution of TKE and argue that TKE advection should be modeled in a closure scheme unless there is an explicit reason to exclude this process. Thus, although our results qualitatively agree with those presented in Larsén and Fischereit (2021), our recommendation to use TKE advection is stronger, based on both performance and physical arguments.

### A3  Evaluation at FINO1 tower with reduced timeframe

3DPBL is still optimal over the MYNN in the FINO1 region when the evaluation timeframe is reduced to match that of the aircraft region. The bias evaluation is consistent with this change. 3DPBL wind speeds and MYNN wind speeds still have comparable biases for all wind farm TKE factors, regardless of the evaluation timeframe (Fig. 13a, Fig. A4a), and the differences in the bias between the PBL cases are still statistically insignificant in both cases (Table 5, Table A3). 3DPBL wind speeds and MYNN wind speeds also still show comparable $EMD$ in both cases (Fig. 13d, Fig. A4d), with statistically insignificant differences (Table 5, Table A3). In contrast, reducing the timeframe for the FINO1 evaluation to match the timeframe for the aircraft observations affects both the PBL $cRMSE$ and PBL $R^2$ evaluations. However, these altered evaluations compensate for each other, and 3DPBL wind speeds still outperform MYNN wind speeds. 3DPBL wind speeds show lower $cRMSE$ than MYNN wind speeds for all wind farm TKE factors (Fig. 13b) when the FINO1 wind speeds are compared over the full timeframe. In contrast, when the FINO1 wind speeds are compared over the reduced timeframe, 3DPBL wind speeds show lower $cRMSE$ than MYNN wind speeds for only a $C_{TKE}$ value of 0 or 0.25 (Fig. A4b). In fact, with a $C_{TKE}$ of 1, 3DPBL wind speeds show higher $cRMSE$ than MYNN wind speeds over this reduced evaluation period. As a consequence, 3DPBL wind speeds are no longer statistically different from MYNN wind speeds according to the $cRMSE$ (Table A3). Shifts also occur with the PBL $R^2$ evaluation when the evaluation timeframe is reduced. Over the full timeframe, 3DPBL wind speeds and MYNN wind speeds show comparable $R^2$ for all wind farm TKE factors (Fig. 13c). However, with the reduced evaluation timeframe, 3DPBL wind speeds consistently show a higher $R^2$ than MYNN wind speeds (Fig. A4c). This difference in $R^2$ is statistically significant (Table A3).

The shifts in the $cRMSE$ and $R^2$ performances with the reduced FINO1 evaluation timeframe diminish – but do not nullify – the statistically significant differences between wind farm TKE factors. Although wind speeds with a $C_{TKE}$ of 0 and a $C_{TKE}$ of 1 show statistically significant different $cRMSE$ when the full evaluation period is considered (Table 5), this statistical significance is not maintained with the reduced evaluation timeframe (Table A3). Similarly, although wind speeds with a $C_{TKE}$ of 0 and $C_{TKE}$ of 1 show statistically significant different $R^2$ values when the full evaluation period is considered





**Figure A4.** Wind speed error metric box plot for the FINO1 tower measurements based on a reduced (14:10:00–16:10:00) timeframe to align with the aircraft measurements. The box and whiskers describe FINO1 model height variability and are based on Q1 (25th percentile), Q3 (75th percentile), and the interquartile range (IQR) (Q3–Q1). The box encloses the IQR, and the whiskers extend to Q1-1.5*IQR and Q3+1.5*IQR. The simulation names are mapped according to the short names provided in Table 4, and the vertical dotted lines visually separate simulations by wind farm TKE factor. (a) Wind speed bias; (b) wind speed $cRMSE$; (c) wind speed $R^2$; (d) wind speed $EMD$.



**Table A3.** $p$ values according to the Mann–Whitney U test. A bold cell indicates statistical significance at $|p| <= 0.05$.

| Parameter | Quantity | Bias | $R^2$ | $EMD$ | $cRMSE$ |
|-----------|----------|------|-------|-------|---------|
| PBL | FINO wind speed | 0.6149 | **0.0** | 0.3786 | 0.4504 |
| $C_{TKE}$ | FINO wind speed | **0.0** | 1.0 | **0.0007** | 0.9817 |

(Table 5), this statistical significance is not maintained with the reduced evaluation timeframe (Table A3). In contrast, both the bias and $EMD$ maintain their statistically significant differences over both evaluation timeframes (Table 5, Table A3). As a consequence, wind speeds with a $C_{TKE}$ of 0 and wind speeds with a $C_{TKE}$ of 1 at the FINO1 site are still statistically

different, although this statistical difference is now limited to two of the four metrics.

## Appendix B: Additional figures

### B0.1 Vertical profiles with all 16 simulations

When we analyzed the model vertical structure of the atmosphere earlier, we identified differences in the atmosphere's vertical structure based on the PBL scheme (Fig. 5). Here, we show that the PBL scheme is still the strongest model determinant of

the atmosphere's vertical structure when we consider all 16 simulations. All 16 simulations preserve the weakly stable surface profile and suggest stronger stable stratification higher above. The MYNN simulations are still warmer and are still more stable near the surface than the 3DPBL simulations, even as we introduce the NWF and no-advection cases (Fig. B1a,d). Some variability exists between the advection and no-advection cases in characterizing the cold-air advection near the surface (Fig. B1b,e); however, these differences are still secondary to differences based on the PBL scheme. NWF cases also have higher

wind speeds than Fitch cases within the turbine rotor region for both sites (Fig. B1c,f), which is consistent with spatial map (Fig. A1e,f), time series (Fig. B3), and flight path (Fig. B6, Fig. B7) analyses. Even among the wind farm cases, MYNN wind speeds are still higher than 3DPBL wind speeds within this portion of the vertical profile (Fig. B1c,f). Importantly, in all cases, the inversions at 500 m are preserved when we consider all 16 simulations (Fig. B1).

### B1 Measurement variability

### B1.1 FINO1 time series for all measurement altitudes

Wind speed differences at the FINO1 tower are consistent through the observational period. Modeled FINO1 Fitch wind speeds are generally restricted within the observational bounds and appropriately capture the temporal shifts throughout the observational period, regardless of the PBL scheme or physics options (Fig. B2). MYNN wind speeds are also consistently higher than 3DPBL wind speeds, consistent with the spatial maps (Fig. 7a). The influence of additional turbulence is also

evident through the wind farm TKE factor. Increasing the wind farm TKE factor decreases the wind speeds, regardless of the PBL scheme (Fig. 8).



**Figure B1.** Potential temperature, wind direction, and wind speed vertical profiles from WRF simulations at a constant latitude of 54.03. In all cases, the solid, black horizontal line indicates the uppermost measurement height, and the dashed horizontal lines indicate turbine hub altitudes for the region. (a) FINO1 potential temperature. (b) FINO1 wind direction. (c) FINO1 horizontal wind speed. (d) Aircraft potential temperature. (e) Aircraft wind direction. (f) Aircraft horizontal wind speed. FINO1 cases are averaged over hours 12:00:00–00:00:00, and the aircraft region cases are averaged over 14:10:00–16:10:00.



**Figure B2.** Time series of modeled horizontal wind speeds compared to FINO1 observations for hours 12:00:00–00:00:00. Both the modeled wind speeds and observed wind speeds are resampled to 30 minutes. (a) 102 m; (b) 91 m; (c) 81 m; (d) 61 m; (e) 51 m; (f) 41 m; (g) 34 m.



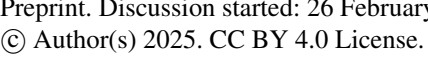

**Figure B3.** Time series of modeled horizontal wind speeds compared to FINO1 observations for hours 12:00:00–00:00:00. Both the modeled wind speeds and observed wind speeds are resampled to 30 minutes for trend evaluation. (a) 102 m; (b) 91 m; (c) 81 m; (d) 61 m; (e) 51 m; (f) 41 m; (g) 34 m.





**Table B1.** FINO1 region wind speed $p$-value reporting. $p_{ind}$ is the $p$ value for the independent samples t-test, $p_{welch}$ is the $p$ value for the Welch test, and $p_{mannwhitney}$ is the $p$ value according to the Mann–Whitney U test. Bold cells indicate statistical significance at $|p| <= 0.05$.

| Parameter | Metric | $p_{ind}$ | $p_{welch}$ | $p_{mannwhitney}$ |
|---|---|---|---|---|
| PBL | BIAS | 0.5598 | 0.5598 | 0.4065 |
| PBL | $R^2$ | 0.4277 | 0.428 | 0.3391 |
| PBL | $EMD$ | 0.7379 | 0.7379 | 0.5629 |
| PBL | $cRMSE$ | **0.0** | **0.0** | **0.0** |
| cTKE | BIAS | **0.0** | **0.0** | **0.0** |
| cTKE | $R^2$ | **0.0013** | **0.0016** | **0.003** |
| cTKE | $EMD$ | **0.0001** | **0.0003** | **0.0007** |
| cTKE | $cRMSE$ | **0.0017** | **0.0019** | **0.003** |

Wind speed differences at the FINO1 tower are consistent through the observational period at all measurement altitudes. Modeled FINO1 Fitch wind speeds share similar shapes and bounds at all measurement altitudes (Fig. B2). Further, MYNN wind speeds are consistently higher than 3DPBL wind speeds, regardless of the measurement altitude (Fig. B2). Increasing the wind farm TKE factor also consistently decreases the wind speeds, regardless of the PBL scheme or the measurement altitude (Fig. B2). NWF cases also consistently have higher wind speeds than Fitch cases at all measurement altitudes, although this difference becomes tighter lower on the tower (Fig. B3). This discrepancy in the Fitch vs. NWF wind speed gap based on the measurement altitude likely reflects the presence (or absence) of an obstructing wind farm. Advection-based differences in wind speed based on the measurement altitude are not evident (Fig. B3).

### B1.2 Aircraft flight path variability for all transects

Trends in the wake-affected region of the flight path are common across transects. 3DPBL TKE is stronger than MYNN TKE, and increasing the wind farm TKE factor further increases the TKE, regardless of the transect (Fig. B4). These TKE-based trends are then mirrored in the wind speed behavior for each transect. Because 3DPBL has higher TKE, 3DPBL also has lower wind speeds (Fig. B5). Further, increasing the wind farm TKE factor further reduces the wind speeds, regardless of the PBL scheme (Fig. B5). As expected, Fitch TKE is higher than NWF TKE (Fig. B7), and NWF wind speeds are consequently higher than Fitch wind speeds (Fig. B6). The gap between Fitch and NWF wind speeds is most pronounced in the middle of the flight path, where the Fitch cases anticipate an obstructing wind farm (Fig. B6). The advection option does not experience any flight path variability (Fig. B6, Fig. B7).

### B2 Statistical significance testing

Statistical significance is maintained regardless of the method of statistical significance testing. At the FINO1 tower, statistical significance for both PBL and $C_{TKE}$ is maintained whether we rely on the Mann–Whitney U test, the independent samples t-test, or the Welch test (Table B1). Similarly, this shared statistical significance across all three tests is maintained within the



**Figure B4.** Aircraft region (250 m) simulated and observed TKE across the flight path. Transect (a) 1; (b) 2; (c) 3; (d) 4; (e) 5; (f) 6.



**Figure B5.** Aircraft region (250 m) simulated and observed horizontal wind speeds across the flight path. Transect (a) 1; (b) 2; (c) 3; (d) 4; (e) 5; (f) 6.



**Figure B6.** Aircraft region (250 m) simulated and observed horizontal wind speeds across the flight path. Transect (a) 1; (b) 2; (c) 3; (d) 4; (e) 5; (f) 6.





**Figure B7.** Aircraft region (250 m) simulated and observed TKE across the flight path. Transect (a) 1; (b) 2; (c) 3; (d) 4; (e) 5; (f) 6.





**Table B2.** Aircraft region wind speed $p$-value reporting. $p_{ind}$ is the $p$ value for the independent samples t-test, $p_{welch}$ is the $p$ value for the Welch test, and $p_{mannwhitney}$ is the $p$ value according to the Mann–Whitney U test. Bold cells indicate statistical significance at $|p| <= 0.05$.

| Parameter | Metric | $p_{ind}$ | $p_{welch}$ | $p_{mannwhitney}$ |
|---|---|---|---|---|
| PBL | BIAS | **0.0045** | **0.0047** | **0.0119** |
| PBL | $R^2$ | **0.0** | **0.0** | **0.0** |
| PBL | $EMD$ | **0.0046** | **0.0057** | **0.0184** |
| PBL | $cRMSE$ | **0.0** | **0.0** | **0.0** |
| cTKE | BIAS | 0.1336 | 0.1336 | 0.1124 |
| cTKE | $R^2$ | 0.8979 | 0.898 | 0.8399 |
| cTKE | $EMD$ | 0.2995 | 0.3004 | 0.2855 |
| cTKE | $cRMSE$ | 0.9422 | 0.9423 | 0.8852 |

**Table B3.** Aircraft region TKE $p$-value reporting. $p_{ind}$ is the $p$ value for the independent samples t-test, $p_{welch}$ is the $p$ value for the Welch test, and $p_{mannwhitney}$ is the $p$ value according to the Mann–Whitney U test. Bold cells indicate statistical significance at $|p| <= 0.05$.

| Parameter | Metric | $p_{ind}$ | $p_{welch}$ | $p_{mannwhitney}$ |
|---|---|---|---|---|
| PBL | BIAS | **0.0** | **0.0** | **0.0** |
| PBL | $R^2$ | 0.995 | 0.995 | 0.937 |
| PBL | $EMD$ | 0.0583 | 0.0588 | 0.1032 |
| PBL | $cRMSE$ | 0.3257 | 0.3261 | 0.3843 |
| cTKE | BIAS | **0.0044** | **0.0044** | **0.0304** |
| cTKE | $R^2$ | **0.0044** | **0.0064** | **0.0194** |
| cTKE | $EMD$ | 0.1267 | 0.1292 | 0.4025 |
| cTKE | $cRMSE$ | **0.0016** | **0.0016** | **0.0017** |

aircraft region, both for wind speed (Table B2) and for TKE (Table B3). Thus, although we report results according to the Mann–Whitney U test because of the small number of simulations per sample and the implied non-normality of each sample,
our results are consistent even if we change our method of statistical significance testing.

## B3   Taylor diagrams

Error metrics were also summarized with Taylor diagrams (Taylor, 2001). Taylor diagrams are polar plots where the standard deviation ($\sigma$) is the radius, the Pearson correlation ($R$) coefficient is described with respect to the azimuth, and the centered root mean square error ($cRMSE$) is a skill score expressed through contours. These visualizations provide a means to simul-
taneously compare data spread, correspondence, and accuracy. These visualizations also identify which model parameters are most important in driving model performance. The $cRMSE$ is a contour-based skill score because of its relation to the other





**Figure B8.** Taylor diagram for modeled wind speeds at the FINO1 tower. The radial distance is proportional to the standard deviation (m s$^{-1}$), the azimuth corresponds to the correlation coefficient (unitless), and the rings represent the centered root mean square error ($cRMSE$) (m s$^{-1}$), where markers in rings closer to the observation's standard deviation (star) have closer agreement than markers in rings farther away. (a) 102 m; (b) 91 m; (c) 81 m; (d) 61 m; (e) 51 m; (f) 41 m; (g) 34 m.



**Figure B9.** Taylor diagram for modeled wind speeds along all six aircraft transects. The radial distance is proportional to the standard deviation (m s$^{-1}$), the azimuth corresponds to the correlation coefficient (unitless), and the rings represent the centered root mean square error ($cRMSE$) (m s$^{-1}$), where markers in rings closer to the observation's standard deviation (star) have closer agreement than markers in rings farther away. Transect (a) 1; (b) 2; (c) 3; (d) 4; (e) 5; (f) 6.



**Figure B10.** Taylor diagram for modeled TKE along the six aircraft transects. The radial distance is proportional to the standard deviation ($m^2$ $s^{-2}$), the azimuth corresponds to the correlation coefficient (unitless), and the rings represent the centered root mean square error ($cRMSE$) ($m^2$ $s^{-2}$), where markers in rings closer to the observation's standard deviation (star) have closer agreement than markers in rings farther away. Transect (a) 1; (b) 2; (c) 3; (d) 4; (e) 5; (f) 6.





**Table B4.** FINO1 region wind speed $p$-value reporting for all simulations. $p_{ind}$ is the $p$ value for the independent samples t-test, $p_{welch}$ is the $p$ value for the Welch test, and $p_{mannwhitney}$ is the $p$ value according to the Mann–Whitney U test. Bold cells indicate statistical significance at $|p| <= 0.05$.

| Parameter | Metric | $p_{ind}$ | $p_{welch}$ | $p_{mannwhitney}$ |
|---|---|---|---|---|
| Advect | BIAS | 0.994 | 0.994 | 0.9466 |
| Advect | $R^2$ | **0.0** | **0.0** | **0.0001** |
| Advect | $EMD$ | 0.9451 | 0.9451 | 0.7248 |
| Advect | $cRMSE$ | 0.0005 | 0.0005 | 0.0005 |
| WF | BIAS | **0.0** | **0.0** | **0.0** |
| WF | $R^2$ | 0.0158 | 0.0387 | 0.0278 |
| WF | $EMD$ | **0.0** | **0.0** | **0.0** |
| WF | $cRMSE$ | **0.0131** | **0.002** | 0.0559 |

**Table B5.** Aircraft region wind speed $p$-value reporting for all simulations. $p_{ind}$ is the $p$ value for the independent samples t-test, $p_{welch}$ is the $p$ value for the Welch test, and $p_{mannwhitney}$ is the $p$ value according to the Mann–Whitney U test. Bold cells indicate statistical significance at $|p| <= 0.05$.

| Parameter | Metric | $p_{ind}$ | $p_{welch}$ | $p_{mannwhitney}$ |
|---|---|---|---|---|
| Advect | BIAS | 0.9892 | 0.9892 | 0.9971 |
| Advect | $R^2$ | 0.6039 | 0.6039 | 0.5553 |
| Advect | $EMD$ | 0.8868 | 0.8868 | 0.8289 |
| Advect | $cRMSE$ | 0.7305 | 0.7305 | 0.6157 |
| WF | BIAS | **0.0** | **0.0** | **0.0** |
| WF | $R^2$ | **0.0** | **0.0** | **0.0** |
| WF | $EMD$ | **0.0** | **0.0** | **0.0** |
| WF | $cRMSE$ | **0.0** | **0.0** | **0.0** |

error metrics:

$$cRMSE = \sqrt{\sigma_p^2 + \sigma_o^2 - 2\sigma_p\sigma_o R} \tag{B1}$$

The PBL scheme is the strongest determinant of model $cRMSE$, whether we consider wind speed at FINO1 (Fig. B8), wind speed measured by the aircraft (Fig. B9), or TKE measured by the aircraft (Fig. B10). The advection option does not suggest $cRMSE$ clustering for either site or comparison variable. Although the wind farm TKE factor does suggest some performance clusters, these clusters are not always consistent. Similarly, wind-farm-option-based clusters are not universal, and some NWF cases outperform Fitch cases. In contrast, the PBL-based clusters are well-defined and consistent across all heights and transects (Fig. B8, Fig. B9, Fig. B10).



**Table B6.** Aircraft region TKE $p$-value reporting for all simulations. $p_{ind}$ is the $p$ value for the independent samples t-test, $p_{welch}$ is the $p$ value for the Welch test, and $p_{mannwhitney}$ is the $p$ value according to the Mann–Whitney U test. Bold cells indicate statistical significance at $|p| <= 0.05$.

| Parameter | Metric | $p_{ind}$ | $p_{welch}$ | $p_{mannwhitney}$ |
|---|---|---|---|---|
| Advect | BIAS | 0.3777 | 0.3777 | 0.3892 |
| Advect | $R^2$ | 0.1195 | 0.1196 | 0.0508 |
| Advect | $EMD$ | 0.8995 | 0.8995 | 0.8061 |
| Advect | $cRMSE$ | 0.1565 | 0.1566 | 0.134 |
| WF | BIAS | **0.0** | **0.0** | **0.0** |
| WF | $R^2$ | **0.0** | **0.0** | **0.0** |
| WF | $EMD$ | **0.0** | **0.0** | **0.0** |
| WF | $cRMSE$ | **0.0** | **0.0** | **0.0** |

This dominant influence of the PBL scheme on meteorological behavior is well-supported by other findings in this work. The potential temperature (Fig. 5a,c) and wind direction (Fig. 5b,d) vertical profiles are influenced by the PBL scheme, but not the advection option or wind farm TKE factor. Simulated wind speeds in the inner domain are also consistently higher under the MYNN scheme than under the 3DPBL scheme outside of the wakes, regardless of the advection option (Fig. 7a). 3DPBL TKE is also consistently higher than MYNN TKE at turbine locations, regardless of the advection option (Fig. 7b).

These PBL-based differences in wind speed and TKE also persist throughout the whole FINO1 observational period (Fig. B2) and across the whole aircraft transect flight path (Fig. B4, Fig. B5).

## B4    Box plots

NWF cases are more consistent in the aircraft region than in the FINO1 region. $R^2$ for NWF wind speeds (Fig. B11c), for example, clearly underperforms $R^2$ for Fitch wind speeds (Fig. B11c) in the aircraft region. Bias and $EMD$ for NWF wind

speeds (Fig. B11a,d) similarly underperform bias and $EMD$ for Fitch wind speeds (Fig. B11a,d). This Fitch vs. NWF distinction is also evident for wind speed $cRMSE$ in the aircraft region, although 3DPBL cases, especially those with a $C_{TKE}$ of 1, compromise this separation (Fig. B11b). Fitch vs. NWF distinctions for TKE in the aircraft region are similar to those for wind speed. NWF TKE underperforms Fitch TKE with respect to $R^2$ (Fig. B12c). In addition, NWF TKE underperforms Fitch TKE with respect to $cRMSE$ (Fig. B12b) and $EMD$ (Fig. B12d), although, again, the 3DPBL cases with a $C_{TKE}$ of 1

threaten this distinction for both metrics. Although the distinction between NWF and Fitch cases is not universal, it is certainly consistent and intuitive. In contrast, NWF cases in the FINO1 region do not show the same level of separation from their Fitch counterparts. $cRMSE$ for FINO1 NWF wind speeds is comparable to that for many FINO1 Fitch wind speeds (Fig. B13b). Similarly, $R^2$ for FINO1 NWF wind speeds is comparable to – and even outperforms – $R^2$ for FINO1 Fitch wind speeds (Fig. B13c). Curiously, this behavior according to the $cRMSE$ (Fig. B13b) and $R^2$ (Fig. B13c) is in contrast to the behavior





**Figure B11.** Error metric box plot for aircraft observations collected at 250 m. The box and whiskers describe aircraft transect variability and are based on Q1 (25th percentile), Q3 (75th percentile), and the interquartile range (IQR) (Q3–Q1). The box encloses the IQR, and the whiskers extend to Q1-1.5*IQR and Q3+1.5*IQR. The simulation names are mapped according to the short names provided in Table 4, and the vertical dotted lines visually separate simulations by wind farm TKE factor. (a) Wind speed bias; (b) wind speed $cRMSE$; (c) wind speed $R^2$; (d) wind speed $EMD$.

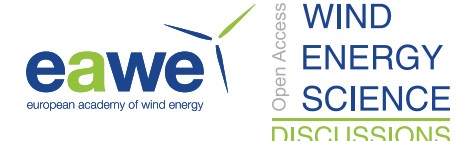

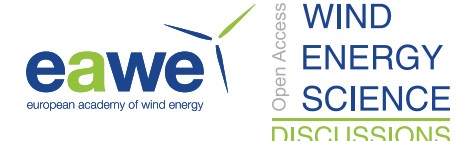

**Figure B12.** Error metric box plot for aircraft observations collected at 250 m. The box and whiskers describe aircraft transect variability and are based on Q1 (25th percentile), Q3 (75th percentile), and the interquartile range (IQR) (Q3–Q1). The box encloses the IQR, and the whiskers extend to Q1-1.5*IQR and Q3+1.5*IQR. The simulation names are mapped according to the short names provided in Table 4, and the vertical dotted lines visually separate simulations by wind farm TKE factor. (a) TKE bias; (b) TKE $cRMSE$; (c) TKE $R^2$; (d) TKE $EMD$.





**Figure B13.** Wind speed error metric box plot for the FINO1 tower measurements. The box and whiskers describe FINO1 model height variability and are based on Q1 (25th percentile), Q3 (75th percentile), and the interquartile range (IQR) (Q3–Q1). The box encloses the IQR, and the whiskers extend to Q1-1.5*IQR and Q3+1.5*IQR. The simulation names are mapped according to the short names provided in Table 4, and the vertical dotted lines visually separate simulations by wind farm TKE factor. (a) Wind speed bias; (b) wind speed $cRMSE$; (c) wind speed $R^2$; (d) wind speed $EMD$.





observed with the bias and $EMD$ in the FINO1 region. In fact, the wind speed bias (Fig. B13a) and $EMD$ (Fig. B13d) both
show a clearer NWF vs. Fitch distinction than any error metric in the aircraft region (Fig. B11, Fig. B12).

Introducing advection helps FINO1 NWF wind speeds outperform FINO1 Fitch wind speeds. Performance does not shift
for either wind speed (Fig. B11) or TKE (Fig. B12) based on the advection option. Similarly, both the bias (Fig. B13a) and
$EMD$ (Fig. B13d) in the FINO1 region are unaffected by the advection option. In contrast, both the $cRMSE$ (Fig. B13b)
and $R^2$ (Fig. B13c) improve by introducing advection. Further, this performance improvement for both error metrics is more
pronounced for NWF cases than for Fitch cases (Fig. B13b,c). Likely the FINO1 site is more sensitive to the advection option
because the FINO1 site is a single grid cell. As such, local TKE shifts are not compensated in error metric averaging as they
are in the broader aircraft region. This interpretation is reinforced by considering differences between the four error metrics.
Although the bias and $EMD$ focus on the center of the distribution, $cRMSE$ and $R^2$ characterize individual locations.

*Author contributions.* JKL conceptualized the project and acquired funding and resources for the project. NJA completed the WRF simulations and carried out the formal analysis and investigation, including developing software and carrying out the visualization, with supervision from JKL, TWJ, and AR. NJA and JKL prepared the initial draft. NJA, JKL, TWJ, and AR reviewed and edited the publication.

*Competing interests.* At least one of the (co-)authors is a member of the editorial board of Wind Energy Science.

*Acknowledgements.* This work was supported by an agreement with NREL under APUP UGA-0-41026-125. This work was authored in
part by the National Renewable Energy Laboratory, operated by Alliance for Sustainable Energy, LLC, for the U.S. Department of Energy
(DOE) under Contract No. DE-AC36-08GO28308. Funding was provided by the U.S. Department of Energy Office of Energy Efficiency
and Renewable Energy Wind Energy Technologies Office and by the National Offshore Wind Research and Development Consortium under
agreement no. CRD-19-16351. The authors acknowledge support from the U.S. Department of Energy (DOE) under DE-EE0009424. The
views expressed in the article do not necessarily represent the views of the DOE or the U.S. Government. The U.S. Government and the
publisher, by accepting the article for publication, acknowledge that the U.S. Government retains a nonexclusive, paid-up, irrevocable,
worldwide license to publish or reproduce the published form of this work, or allow others to do so, for U.S. Government purposes. Neither
NYSERDA nor OceanTech Services/DNV have reviewed the information contained herein, and the opinions in this report do not necessarily
reflect those of any of these parties. Data storage supported by the University of Colorado Boulder 'PetaLibrary'. This work utilized the
Alpine high-performance-computing resource at the University of Colorado Boulder. Alpine is jointly funded by the University of Colorado
Boulder, the University of Colorado Anschutz, and Colorado State University. A portion of this research was performed using computational
resources sponsored by the DOE's Office of Energy Efficiency and Renewable Energy and located at NREL. Author TWJ is grateful for
support in part from the U.S. Department of Energy Wind Energy Technologies Office through Contract DE-A05-76RL01830 to Pacific
Northwest National Laboratory (PNNL). The U.S. National Science Foundation National Center for Atmospheric Research is a subcontractor
to PNNL under Contract 659135. The National Center for Atmospheric Research is a major facility sponsored by the U.S. National Science
Foundation under Cooperative Agreement No. 1852977.





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
