# Peer review of "A North Sea in situ evaluation of the Fitch Wind Farm Parametrization within the Mellor–Yamada–Nakanishi–Niino and 3D Planetary Boundary Layer schemes"

_Wind Energy Science, 2025_

## Referee Comment (RC1)

**Review of "A North Sea in situ evaluation of the FitchWind Farm Parametrization within the Mellor–Yamada–Nakanishi–Niino and 3D Planetary Boundary Layer schemes" by Agarwal et al.**

The manuscript compares a new planetary boundary layer scheme to the standard one used for wind farm modeling in WRF. It evaluates the accuracy of the PBL schemes with offshore measurements of two independent sources. A thorough analysis is carried out with respect to model setup and statical metrics.

This work addresses an understudied, yet extremely relevant aspect of mesoscale wind farm modeling. However, several aspects of the manuscript need major improvement before acceptance.

**Specific comments**

- The authors should significantly narrow the scope of the work to make the analysis more comprehensible and easier to read. Since the main focus is to compare 3DPBL with MYNN, all results not directly supporting this comparison should be removed from the analysis. Specifically:
    - Varying the TKE factor is a relevant analysis, but clouds the clarity of comparison between PBL models. Simply choose one TKE factor (suggestion: default 0.25) and present only these results in figures 5-15. Then add a final section at the end of the results that focuses on the effect of the TKE factor, illustrated by one figure.
    - The appendix should serve the manuscript by providing extra explanations or results that didn't fit to the story, whereas currently the appendix feels more like a 'result dump'. It should be revised what results are interesting enough to add to the appendix.
- The conditions on which the evaluation takes place should be more clearly described. Currently, the combination of four statistical metrics, three data sets (or more if you separate between FINO heights) and six WRF runs (2 PBLs times 3 TKE factors) is extremely overwhelming when trying to find the main result. Especially since the results are not very consistent, this results in a very chaotic discussion and conclusion, see for instance the abstract and l. 316-324. You claim that the 3DPBL is better than MYNN in the rotor area (l. 344-346), which is not that clear from the results shown before.
- The differences between the two PBL schemes should be related to the physics that are modeled in these schemes. In the discussion of the results, it is currently mostly stated that there is a difference, but an interpretation on why this makes sense is missing. Besides, a paper comparing different PBL schemes should definitely include a section in the methodology that briefly described the fundamental principles of these schemes, possibly supported by the most relevant equations. This can then be referred to explain the results. Some examples:
    - L. 235-236: "MYNN simulations … the surface."
    - L. 249-250: "Both the … simulation (Fig. 6a,c)."
    - L. 261-263: "MYNN average … wind speeds."
    - L. 294-295: "TKE in … MYNN simulations."

- The manuscript draws some very thin conclusions that are not (fully) supported by the results or explained in the accompanying text. Some examples:
    - l. 268-270: You say that the wind speed in the wakes in MYNN are stronger because the higher inflow wind speed. However, the difference is very large (up to 1 m/s). I suspect there is more to it than just the difference in inflow wind speed, especially since Fig. 7a shows that the difference in WS offshore are not that large. To proof your claim, I'd recommend comparing the wakes under the same inflow wind speed. This can be done by either running WRF over a longer period of time and binning the results, or designated idealized WRF runs. This would help greatly in explaining the remainder of your results: is the difference between the PBL schemes due to a change in estimated background wind speed, or because of wake dissipation?
    - L. 344-346: You claim that your results support that 3DPBL is better than MYNN in the rotor area, although the results of FINO1 show that this is highly dependent on the statistical metric used.
    - L. 399-406: You claim that a TKE factor of 1 is better for MYNN in your study. However, to claim this you need to be absolutely sure that the background (without turbines) wind speed and TKE are correct. It could be possible underestimates WS and TKE, which are now corrected by adding more turbulence. This doesn't mean that this is physically correct. Besides, Fig. 13-15 clearly shows that for both PBL schemes the trend is the same, there is just a systematic offset. As the TKE factor represent a physical process, it is wrong to conclude that one TKE factor is better for MYNN and other for 3DPBL.
- A discussion on the uncertainty of your results is completely absent. A critical evaluation of your results to the uncertainty in the background wind speed is essential for the interpretation of the results. You show that the background wind speed is about 1 m/s higher in MYNN, but then only evaluate on the waked wind speed. Is 3DPLB then really better, or is it just cancelling out errors?
Additionally, please address the effect of model grid resolution and generalizability to other sites and longer time series.

**Technical corrections**

- l. 45-46: "To date" is followed by a reference to a paper from 2014. Update this
- l. 53-65: This section states what other papers have carried out but does not discuss their findings.
- l. 72: somewhere here the 3D TKE scheme should also be mentioned, and compared (qualitatively) to 3D PBL.
- L. 82-87: It is unclear how this section contributes to the storyline in the introduction
- Fig. 2: ensure that the axes are equal, which should result in square grid boxes.
- L. 130-131: mention exactly what WRF version
- L. 143: "[]" should be "[-]"
- L. 153: detail like file name not needed

- Fig. 4: added benefit of showing this figure unclear
- L. 187-188: Do these studies use the same grid size as you do? Or is the 2km they recommend based on their grid resolution?
- L. 217-219: provide equations for EMD like done for all other metrics
- L. 231-232: add Introduction to help the reader prepare for what's coming
- L. 233: header not really appropriate for this section
- L. 237: wind direction in plot is shown to rotate from southwest to west, so it veers as is expected in SBL
- Fig. 5: why are no measurement added here? Or alternatively, why not use the NWF runs for a cleaner analysis of the profiles?
- L. 244: Add PBLH as reference
- Fig. 6: why are there only contour plots for TKE, and not for WS? Suggest to either add WS contour plots, or just add TKE profile to Fig. 5.
- Sect. 3.2.1: Why is there not TKE/TI comparison for FINO1?
- L. 281-282: increasing TKE factor increases wind speed
- L. 287-288: what should be noted is that the wind speeds a lower altitudes are also lower, resulting in lower biases. Alternatively look at relative biases
- Fig. 8: unclear what the mean values are
- Fig. 11: relate findings to FINO results at multiple heights: can these results be extrapolated to arrive at the results shown here? Additionally, 11c shows the opposite to FINO, please explain why.
- L. 307: The issue of relative sensor alignment is very relevant and deserves more attention than this single sentence. Mention this in the methods section when introducing the aircraft measurements and include it in the discussion of uncertainties.
- L. 355: Fig. 14a is aircraft, but text says FINO1.
- L. 363: Provide references for "recent scientific discussion"
- L. 433-447: This outlook does not belong to the conclusion section.
- L. 461-462: If you claim this, show that the 3DPLB scheme indeed extracts more momentum for higher altitudes.
- L. 509-523: Larsen & Fishereit is mentioned 7 times in just a few sentences. Rewrite this. This happens all over the manuscript, please review that you don't insert unnecessary references.
- Appendix A3: You show here that the results shown in the main body of the manuscript hold for just the 2h aircraft data is available. If this is true, why show the 12h analysis at all? Explain in the main body of the text why you argue that the 12h analysis is better. Additionally, FINO1 allows for an analysis at a much longer time scale. Justify why you restrict your analysis to just 12 hours.

---

## Author Comment (AC1)

* * *
*Editor's and reviewers' comments appear in italics*; **our responses appear in boldface blue text**.

**Reviewer 1**

***Review of "A North Sea in situ evaluation of the FitchWind Farm Parametrization within the Mellor–Yamada–Nakanishi–Niino and 3D Planetary Boundary Layer schemes" by Agarwal et al.*** *The manuscript compares a new planetary boundary layer scheme to the standard one used for wind farm modeling in WRF. It evaluates the accuracy of the PBL schemes with offshore measurements of two independent sources. A thorough analysis is carried out with respect to model setup and statical metrics.*
*This work addresses an understudied, yet extremely relevant aspect of mesoscale wind farm modeling. However, several aspects of the manuscript need major improvement before acceptance.*

**We thank the reviewer for their thoughtful assessment as well as their acknowledgement of the importance of the content covered in this manuscript. We feel that the above accurately characterizes the scope of our manuscript.**

*Specific comments*

- *The authors should significantly narrow the scope of the work to make the analysis more comprehensible and easier to read. Since the main focus is to compare 3DPBL with MYNN, all results not directly supporting this comparison should be removed from the analysis. Specifically: Varying the TKE factor is a relevant analysis, but clouds the clarity of comparison between PBL models. Simply choose one TKE factor (suggestion: default 0.25) and present only these results in figures 5-15. Then add a final section at the end of the results that focuses on the effect of the TKE factor, illustrated by one figure.*

**We thank the reviewer for their helpful suggestion, which was also echoed by Reviewer #2. We have revised the manuscript to use the default 0.25 wind farm TKE factor and have moved the other TKE options to Sect. A2 of the Appendix.**

- *The appendix should serve the manuscript by providing extra explanations or results that didn't fit to the story, whereas currently the appendix feels more like a 'result dump'. It should be revised what results are interesting enough to add to the appendix.*

We thank the reviewer for this thoughtful guidance, which is also echoed by Reviewer 2. Based on this joint guidance, we have revised the Appendix in the following method to make it more of a story by doing the following:

- We have removed the entirety of Appendix B from the text
- We have preserved the NWF vs. WF distinction as part of the appendix to better distinguish the differences between PBL schemes in "baseline" conditions (Sect A1)
- We have moved the discussion of the effect of the wind farm TKE factor to the Appendix (Sect A2)
- We have moved the advection discussion to the Appendix (Sect A3)

- *The conditions on which the evaluation takes place should be more clearly described.*

We thank the reviewer for their thoughtful comment. We have expanded upon the conditions under which the evaluation took place by including vertical profiles (new Fig. 4 and Fig. 5) with the observations to describe the atmospheric conditions at both sites. We start this discussion on updated line 350:

"The modeled atmospheric stability at both the FINO1 location (Fig. 4a,b) and over the aircraft region (Fig. 4f,g) suggests a weakly stable profile near the surface with stronger stability aloft. MYNN simulations are slightly warmer (according to potential temperature) and more stable near the surface than 3DPBL simulations (Fig. 4b,g). For both sites, these modeled temperature profiles are consistent with available observations (Fig. 4a,f) and the observed potential temperature in the aircraft region is slightly warmer than the modeled potential temperature (Fig. 4g). Because the modeled air temperatures are almost identical between models (Fig. 4a,f), these slight differences in the surface stability could be a consequence of the greater TKE with the 3DPBL scheme that encourages slightly more mixing…"

This discussion then extends to the aircraft profile flights starting on updated line 368:

"The aircraft profile flights suggest a similar vertical structure to that for the FINO1 and aircraft transect regions. Both MYNN and 3DPBL temperature and potential temperature profiles at the locations of the profiles (Fig. 5a,b) suggest a weakly stable profile near the surface with stronger stability aloft. This stability is reinforced by the aircraft profile observations (Fig. 5f,g). A LLJ again emerges both in wind speed (Fig. 5d,i) and TKE (Fig. 5e,j) maxima. Finally, both modeled (Fig. 5c) and observed (Fig. 5h) wind direction from the aircraft profiles again suggest warm-air advection with an inversion that is supported by the modeled PBL height.

For the aircraft profile flights, the vertical structure suggests slight discrepancies between model and observation both in terms of values and shape (Fig. 5)..."

[Figure]

**Figure 4. Temperature, potential temperature, wind direction, wind speed, and TKE vertical profiles from observations and WRF simulations for both sites. Dashed lines indicate the modeled PBL height and the grey region indicates the turbine rotor region. Observations in the aircraft region are separated between and even and odd transects. (a) FINO1 temperature; (b) FINO1 potential temperature; (c) FINO1 wind direction; (d) FINO1 horizontal wind speed; (e) FINO1 TKE; (f) aircraft temperature; (g) aircraft potential temperature; (h) aircraft wind direction; (i) aircraft horizontal wind speed; (j) aircraft TKE. FINO1 cases are averaged over hours 12:00:00–00:00:00 and the aircraft region cases are averaged over 14:10:00–16:10:00. FINO1 TKE calculations based on observations were not available due to the coarse temporal resolution of the wind speeds. FINO1 potential temperature calculations based on observations were not available due to a lack of pressure observations.**

[Figure]

**Figure 5. Observed and modeled vertical profiles for the aircraft vertical profile flights (Table 3, Fig. 1b). In all cases, the horizontal dashed line indicates the modeled PBL height, and the color differentiates the PBL scheme. The top row of panels corresponds to modeled output and the bottom row of panels corresponds to the aircraft profile observations. Modeled output are determined to be a given middle cell for each profile as in Larsén and Fischereit (2021) based on the timestep indicated in Table 3. (a) modeled temperature; (b) modeled potential temperature; (c) modeled wind direction; (d) modeled horizontal wind speed; (e) modeled TKE; (f) observed temperature; (g) observed potential temperature; (h) observed wind direction; (i) observed horizontal wind speed; (j) observed TKE.**

- *Currently, the combination of four statistical metrics, three data sets (or more if you separate between FINO heights) and six WRF runs (2 PBLs times 3 TKE factors) is extremely overwhelming when trying to find the main result. Especially since the results are not very consistent, this results in a very chaotic discussion and conclusion, see for instance the abstract and l. 316-324.*

We acknowledge the reviewer's concern over the magnitude of the scope of presented results. We have refined the discussion by moving the wind farm TKE factor discussion to the Appendix (Sect. A2) and focusing results on the distinction between PBL schemes.

- *You claim that the 3DPBL is better than MYNN in the rotor area (l. 344-346), which is not that clear from the results shown before. The differences between the two PBL schemes should be related to the physics that are modeled in these schemes. In the discussion of the results, it is currently mostly stated that there is a difference, but an interpretation on why this makes sense is missing. Besides, a paper comparing different PBL schemes should definitely include a section in the methodology that briefly described the fundamental principles of these schemes, possibly supported by the most relevant equations. This can then be referred to explain the results.*

We thank the reviewer for their thoughtful critique. Reviewer 2 similarly expressed a desire for a section that outlines the physical differences between the two models. We have included a new subsection of the methods section (the new Sect. 2.1) that describes these differences and references the fundamental governing equations for each model. Notably, these two models approach the same level 2.5 closure equations with different horizontal turbulent mixing, length scales, and corresponding closure constants. Based on these differences in horizontal turbulent mixing, length scales, and closure constants, we believe that consequent differences in TKE characterization drive observed differences in other meteorological features such as surface stability strength and wind speed. We address the concern by providing a section that explains the fundamental differences between the models (the new Sect. 2.1), as well as introducing a new figure (Fig. 7), referencing the idealized analysis performed in Rybchuk et al. (2022), and providing interpretations that are consistent with other results in this work.

- *Some examples: L. 235-236: "MYNN simulations … the surface."*

In this section, now starting on line 351, we argue that the slight differences in stability are a consequence of the increased TKE from the 3DPBL scheme. This increased TKE with the 3DPBL scheme encourages slightly more mixing:

"Because the modeled air temperatures are almost identical between models (Fig. 4a,f), these slight differences in the surface stability could be a consequence of the greater TKE with the 3DPBL scheme that encourages slightly more mixing"

- *L. 249-250: "Both the … simulation (Fig. 6a,c)."*

In this section, now starting on line 391, we point out that the larger TKE with the 3DPBL scheme in stable conditions, which is consistent with the idealized, stable conditions simulated in Rybchuk et al. (2022), reflects the fundamental differences between the 3DPBL and MYNN models:

"These differences in TKE between the PBL schemes, which are consistent with the idealized, stable conditions simulated in Rybchuk et al. (2022), reflect the fundamental differences between the models. Notably, while the MYNN scheme uses Smagorinsky mixing to characterize horizontal turbulent mixing, the 3DPBL scheme instead calculates the horizontal turbulent flux divergences explicitly. The two models also rely on different length scales and empirical constants"

- L. 261-263: "MYNN average … wind speeds."

In this section, now starting on line 416, we maintain our interpretation that larger TKE with the 3DPBL scheme implies greater momentum extraction that results in reduced wind speeds.

"This larger TKE with the 3DPBL scheme extracts more momentum from the mean wind, resulting in a greater reduction in wind speed. This finding that MYNN wind speeds are faster than 3DPBL wind speeds is consistent with other comparisons of these two PBL schemes, completed in both real and idealized conditions (Juliano et al., 2022; Rybchuk et al., 2022; Arthur et al., 2022; Peña et al., 2023; Arthur et al., 2024)."

We also introduce a new Fig. 7, consistent with the other reviewer's suggestion, to document the differences in vertical structure of wind speeds between the two PBL schemes to support our claim.

[Figure]

**Figure 7. Modeled wind speed cross-section at a constant latitude of 54.03. (a) FINO1 3fa_025; FINO1 mfa_025; (e) FINO1 3fa_025- mfa_025; (b) aircraft 3fa_025; (d) aircraft mfa_025; (f) aircraft 3fa_025 - mfa_025. The horizontal dashed black line denotes the average modeled PBL height, the star indicates the FINO1 tower location, the "X" marks the first transect path, and the black circles indicate the turbine hub height.**

**Finally, we corroborate this interpretation in Sect. A2 of the Appendix. Fig. A3a of Sect. A2 demonstrates how increasing the amount of TKE (in this case, via the wind farm TKE factor) reduces the wind speeds.**

[Figure]

**Figure A3. Error metric box plot for aircraft observations collected at 250 m for wind speed. The box and whiskers describe aircraft transect variability and are based on Q1 (25th percentile), Q3 (75th percentile), and the interquartile range (IQR) (Q3–Q1). The box encloses the IQR, and the whiskers extend to Q1-1.5\*IQR and Q3+1.5\*IQR. The simulation names are mapped according to the short names provided in Table 6, and the vertical dotted lines visually separate simulations by wind farm TKE factor. (a) Wind speed bias; (b) wind speed cRMSE; (c) wind speed R2; (d) wind speed EMD.**

- *L. 294-295: "TKE in … MYNN simulations."*

In this section, now starting on line 449, we reference the figure from the discussion in the previous bullet point and maintain the same interpretation.

"As noted earlier, these differences in TKE between the PBL schemes reflect the fundamental differences between the models in length scales, empirical constants, and horizontal mixing approaches."

- *The manuscript draws some very thin conclusions that are not (fully) supported by the results or explained in the accompanying text. Some examples: l. 268-270: You say that the wind speed in the wakes in MYNN are stronger because the higher inflow wind speed. However, the difference is very large (up to 1 m/s). I suspect there is more to it than just the difference in inflow wind speed, especially since Fig. 7a shows that the difference in WS offshore are not that large. To proof your claim, I'd recommend comparing the wakes under the same inflow wind speed. This can be done by either running WRF over a longer period of time and binning the results, or designated idealized WRF runs. This would help greatly in explaining the remainder of your results: is the difference between the PBL schemes due to a change in estimated background wind speed, or because of wake dissipation?*

Thank you for the suggestion to use idealized simulations to explore the differences between MYNN and 3DPBL. The design, execution, and analysis of idealized runs to explore this question has already been conducted and discussed in Rybchuk et al. (2022), who show that different geostrophic forcing is required for MYNN and 3DPBL to generate the same hub-height wind speeds. The abstract of the discussion paper (https://wes.copernicus.org/preprints/wes-2021-127/wes-2021-127.pdf) directly addresses this concern: "For these idealized scenarios, MYNN consistently predicts internal wakes that are 0.25–1.5 m s$^{-1}$ stronger than internal 3DPBL wakes. However, because MYNN predicts stronger inflow winds than 3DPBL, MYNN predicts average capacity factors that are as large as 13 percentage points higher than with the 3DPBL, depending on the stability." On updated line 419, we acknowledge that:

"This finding that MYNN wind speeds are faster than 3DPBL wind speeds is consistent with other comparisons of these two PBL schemes, completed in both real and idealized conditions (Juliano et al., 2022; Rybchuk et al., 2022; Arthur et al., 2022; Peña et al., 2023; Arthur et al., 2024)."

- *L. 344-346: You claim that your results support that 3DPBL is better than MYNN in the rotor area, although the results of FINO1 show that this is highly dependent on the statistical metric used.*

The reviewer is thoughtful to acknowledge that the "optimal PBL scheme" depends on the definition of "optimal." This sentiment is also expressed by Reviewer 2. Here, we alter

**the sentence, now starting on line 507, to specify "Our cRMSE results in the turbine rotor layer" as opposed to "Our results in the turbine rotor layer".**

- *L. 399-406: You claim that a TKE factor of 1 is better for MYNN in your study. However, to claim this you need to be absolutely sure that the background (without turbines) wind speed and TKE are correct. It could be possible underestimates WS and TKE, which are now corrected by adding more turbulence. This doesn't mean that this is physically correct. Besides, Fig. 13-15 clearly shows that for both PBL schemes the trend is the same, there is just a systematic offset. As the TKE factor represent a physical process, it is wrong to conclude that one TKE factor is better for MYNN and other for 3DPBL.*

**Thank you, we agree that varying the TKE factor induces a trend that is distracting from the overall results. Due to this behavior and other reviewer comments, we have decided to keep the TKE factor constant between the two schemes and present only the TKE factor of 0.25 in the main text; the other factors are included in Sect. A2 of the Appendix. We also reframe our discussion of the wind farm TKE factors in Sect. A2 of the Appendix to instead emphasize how turbulence overestimation can inadvertently improve wind speed biases.**

- *A discussion on the uncertainty of your results is completely absent. A critical evaluation of your results to the uncertainty in the background wind speed is essential for the interpretation of the results. You show that the background wind speed is about 1 m/s higher in MYNN, but then only evaluate on the waked wind speed. Is 3DPLB then really better, or is it just cancelling out errors? Additionally, please address the effect of model grid resolution and generalizability to other sites and longer time series.*

**We understand the reviewer would like more discussion of uncertainty with respect to many considerations. We have now incorporated this discussion and itemize our extensive revisions here:**

- **We address the uncertainty of our results in background wind speed by discussing the differences found in the inflow wind speeds between the two PBL schemes according to the Rybchuk et al., 2022 idealized runs under stable, neutral, and unstable conditions. Starting on line 419, we state:**

**"This finding that MYNN wind speeds are faster than 3DPBL wind speeds is consistent with other comparisons of these two PBL schemes, completed in both real and idealized conditions (Juliano et al., 2022; Rybchuk et al., 2022; Arthur et al., 2022; Peña et al., 2023; Arthur et al., 2024). This finding is also documented further in Fig. 7 and in Sect. A1."**

- **We address the uncertainty in the model grid cell spacing by addressing the works of Peña et al., 2023, and Arthur et al., 2024. Both studies compare the MYNN and 3DPBL schemes with a smaller grid cell spacing and arrive at differing conclusions with respect to which PBL scheme is optimal. Starting on line 517, we state:**

"Grid cell spacing may influence the PBL comparison in this work. Notably, the 3DPBL is theorized to improve model performance in the "terra incognita", where the NWP horizontal grid spacing $\Delta x$ approaches a similar magnitude to the PBL depth z (Wyngaard, 2004). Indeed, Arthur et al. (2022) found that the 3DPBL scheme outperformed the MYNN scheme in their simulations with a large-eddy simulation (LES) grid that has a finer cell spacing than that used in our mesoscale simulations. Thus, the mesoscale grid cell spacing necessary for this analysis, while vital to contextualize the results of this work with the broader literature, may restrict the potential benefits of the 3DPBL scheme over the MYNN scheme. At the same time, Peña et al. (2023) compared 3DPBL and MYNN to LES output and found that MYNN outperformed 3DPBL when the finest grid cell spacing was used. Thus, grid cell spacing may influence–but not necessarily dictate–the optimal PBL scheme."

-   We address the generalizability to other sites in the Conclusion as an opportunity for future work. Because field campaigns like the Third Wind Farm Improvement Project (WFIP3) are still actively collecting data, we see an analysis of data for longer-term field campaigns like WFIP3 as a natural next extension of this work. Starting on line 557:

"Subsequent investigations could explore other case studies to provide perspective into the generalizability of the results across other sites. Similarly, datasets from the third Wind Forecast Improvement Project (WFIP3) could be useful to explore how offshore wind characterization might differ between the North Sea and the eastern United States (WFIP3). Moreover, datasets from the land-based, horizontally homogeneous American WAKE experimeNt – or AWAKEN – campaign (Moriarty et al., 2024) could be useful to study because previous land-based studies analyzing the 3DPBL scheme have involved complex terrain and far fewer detailed observations."

-   We address the uncertainty of the time series length by emphasizing the influence of stability conditions on the PBL comparison (Rybchuk et al., 2022). By restricting the FINO1 analysis to the 12 hours present in the Ali et al., 2023 analysis, we allow the analysis present in the Agarwal et al. (2025) manuscript to be contextualized in the broader literature; avoid differences in stability conditions that could influence our PBL comparison; and target the stability conditions that contribute both the strongest and longest wakes. Starting on line 531, we state:

"The stable stratification present in this case study also improves the utility of the results of this PBL comparison. By restricting this analysis to time periods considered in previous analyses for this case study, not only are the results contextualized within the broader literature, but the conditions that contribute the strongest and longest wakes are also highlighted. Thus, while other analyses of this region may approach the lack of available in situ observations by introducing statistical downscaling methods to explore scientific questions around diurnal, seasonal, and climatic trends (Fischereit et al.,

**2022b), this analysis instead addresses scientific questions that are best-suited with in situ observations alone.”**

**Technical corrections**

- *l. 45-46: "To date" is followed by a reference to a paper from 2014. Update this*

**We thank the reviewer for this comment. Reviewer 2 similarly acknowledged that a citation from 2014 was not representative of the most current advancements in wind farm parameterization validations. We have updated this reference to a more recent, highly-cited review of wind farm parameterization validations (Fischereit et al., 2022).**

- *l. 53-65: This section states what other papers have carried out but does not discuss their findings.*

**We thank the reviewer for this comment. We have added a more detailed discussion of each of the relevant analyses of this case study and this discussion includes key findings from each work. This discussion now begins on line 49 and extends to line 75:**

**"A recent North Sea measurement campaign has stimulated interest in WFP intercomparison and validation efforts. The Wind Park Far Field (WIPAFF) project was an aircraft expedition to understand offshore wind wake behavior in the German Bight. This expedition took place in a location with multiple wind farms, with 41 total flights between 6 September 2016 and 15 October 2017 (Platis et al., 2018; Bärfuss et al., 2021). Siedersleben et al. (2018b) modeled a case study that included one day of these aircraft observations and found that, because of challenges in characterizing air-sea interactions with a mesoscale model, improving background flow characterization contributed greater model improvement than any wind farm-specific parameter configuration. Siedersleben et al. (2018a) then extended the analysis of this one case study to demonstrate the presence of hub-height potential temperature and water vapor wakes during stably-stratified conditions. Siedersleben et al. (2020) then leveraged 3 days of these aircraft observations that occurred during stable conditions to explore the sensitivity of grid cell spacing and TKE advection within the Fitch parameterization. Siedersleben et al. (2020) found that refined horizontal and vertical grid cell spacing improved model agreement and that the Fitch et al. (2012) scheme improved performance with a non-zero wind farm TKE factor. Siedersleben et al. (2020) also found (with the bug-advected Fitch et al. (2012) scheme) that model performance improved when the advection option was turned off. Larsén and Fischereit (2021) extended the work of Siedersleben et al. (2020) by comparing the explicit wake parameterization (EWP) (Volker et al., 2015) and Fitch (Fitch et al., 2012) schemes and exploring model performance during wind farm interactions with low-level jets (LLJs). Larsén and Fischereit (2021) found that the EWP significantly underestimated TKE compared to the Fitch scheme and that the wind farm TKE factor within the Fitch et al. (2012) scheme was the most sensitive modeling parameter. Larsén and Fischereit (2021) also noted that introducing wind farms into the**

domain altered the LLJ profile. Larsén et al. (2024) then extended the work in Larsén and Fischereit (2021) by introducing ocean and wave coupling. Larsén et al. (2024) found that wind farm wakes reduced surface wind speeds and wave heights, except for when the wind farm TKE factor was 1. When the wind farm TKE factor was 1, excessive TKE generation instead led to increased surface wind speeds and wave heights. Ali et al. (2023) also used data from one day of the Siedersleben et al. (2020) case study to validate five (Fitch et al., 2012; Volker et al., 2015; Abkar and Porté-Agel, 2015; Pan and Archer 2018; Redfern et al., 2019) common WFPs with aircraft measurements as well as nearby meteorological tower and synthetic aperture radar observations. Ali et al. (2023) found that that the EWP (Volker et al., 2015) underestimated TKE as compared to the Fitch et al. (2012) scheme and that while the Redfern et al. (2019) parameterization did not noticeably deviate from the Fitch et al. (2012) scheme, the Pan and Archer (2018) parameterization showed a significant drop in power generation and the Abkar and Porté-Agel (2015) parameterization predicted lower levels of TKE."

- *l. 72: somewhere here the 3D TKE scheme should also be mentioned, and compared (qualitatively) to 3D PBL.*

We thank the reviewer for their request. We now include a discussion of 3DTKE in our new methods subsection that details the physical differences between MYNN and 3DPBL. This section, Sect. 2.1, can be found starting on line 100 and includes a qualitative comparison of MYNN, 3DPBL, and 3DTKE, and continues on to a derivation of the governing equations for MYNN and 3DPBL:

"The analysis in this work compares the MYNN 2.5 (Mellor and Yamada, 1982; Olson et al., 2019) and NCAR 3DPBL (Juliano et al., 2022) schemes. A brief description of the differences between these two schemes is provided here, while a more complete description of the differences between the 3DPBL and MYNN schemes can be found in Kosović et al. (2020), Juliano et al. (2022), and Rybchuk et al. (2022). (Note the 3DPBL scheme described in this work is not to be confused with 3DTKE (Zhang et al., 2018). 3DTKE is a scale-adaptive model that relies on a level 3 closure. More information about 3DTKE can be found in Zhang et al. (2018).)"

- *L. 82-87: It is unclear how this section contributes to the storyline in the introduction*

Thank you. Reviewer 2 has similarly expressed a concern regarding the added value of this paragraph. We have removed this paragraph from the text.

- *Fig. 2: ensure that the axes are equal, which should result in square grid boxes.*

Thank you, we have updated Fig. 2 to ensure that the axes are equal.

[Figure]

**Figure 2. Map of the number of turbines per WRF grid cell in the innermost domain of the 1.67 km resolution for the two regions of interest within the inner region. The two axes represent the WRF grid system. (a) FINO1 site, with the tower marked with a star. (b) Aircraft site, with the six transect paths traced in black.**

- *L. 130-131: mention exactly what WRF version*

**Thank you, we now note that we used WRFV4.4.2 on line 206.**

- *L. 143: "[]" should be "[-]"*

**Thank you, we have updated (now) line 218 to reflect this change.**

- L. 153: detail like file name not needed

**Thank you, we have removed the file name details. The section, now starting on line 258, reads "...through turbine specification files.." and "with a file from Ali et al.".**

- *Fig. 4: added benefit of showing this figure unclear*

**We thank the reviewer for this comment. We agree that each figure should be intentional. Because Reviewer 2 has asked that this figure be included and that the curves be labeled, we propose that one way to address these joint concerns is to combine this figure with the previous power curve figure with each of the curves labeled as (updated) Fig. 3. We believe that this solution better contextualizes the purpose of the figure, improves the utility for researchers interested in turbine-specific characteristics, and simultaneously de-emphasizes this figure as a standalone component of the narrative.**

[Figure]

**Figure 3. (a) Curve illustrating turbine $C_T$, $C_P$, and power specifications for the turbine model in the Gode wind farm. (b) Drag proxy for each of the eight turbine models present in this case study.**

- *L. 187-188: Do these studies use the same grid size as you do? Or is the 2km they recommend based on their grid resolution?*

**We thank the reviewer for the close attention to our data processing. Both Platis et al., 2018 and Larsén & Fischereit, 2021 recommend a window between 1.5-2 km and report that results are consistent regardless of whether a window of 1.5 km or 2 km is employed. The 1.67 km model grid cell spacing in this work is also consistent with that of prior studies, including Ali et al., 2023 and Siedersleben et al., 2020. Larsén & Fischereit, 2021 use a 2 km grid cell spacing.**

- *L. 217-219: provide equations for EMD like done for all other metrics*

**Thank you, we have introduced the form of the EMD equation consistent with the python package that we use for our analysis on (updated) line 333.**

$$EMD(u,v) = \inf_{\pi \in \Gamma(u,v)} \int_{\mathbb{R} \times \mathbb{R}} |x - y| d\pi(x,y) \qquad (23)$$

where $\Gamma(u,v)$ is the set of probability distributions on $\mathbb{R} \times \mathbb{R}$ whose marginals are $u$ and $v$ on the first and second factors, such that $u(x)$ is the probability of $u$ at position $x$ and $v(x)$ is the probability of $v$ at position $x$ (Ramdas et al., 2015). Here, the $EMD$ was calculated with the Python function wasserstein_distance() from the SciPy library (Virtanen et al., 2020).

- *L. 231-232: add Introduction to help the reader prepare for what's coming*

**Thank you, we have added a short introduction to the Results section that reads, now starting on line 345:**

**"We outline the results of the 3DPBL and MYNN evaluation below. We begin by discussing differences in overall site characterization, then move to a consideration of systematic influences on model performance, and finish with a statistical evaluation of model performance at both sites."**

- *L. 233: header not really appropriate for this section*

**Thank you, we instead propose the title "Site analysis" instead of "Site characterization" on (updated) line 348.**

- *L. 237: wind direction in plot is shown to rotate from southwest to west, so it veers as is expected in SBL*

**Thank you, we alter this line, now line 356, to read "The wind direction profiles at both FINO1 (Fig. 4c) and the aircraft transect (Fig. 4h) regions show veering wind (i.e., the wind direction rotates from southwesterly direction near the surface to westerly aloft), suggesting warm-air advection."**

- *Fig. 5: why are no measurement added here? Or alternatively, why not use the NWF runs for a cleaner analysis of the profiles?*

**We thank the reviewer for their suggestion. Reviewer 2 similarly suggested that we validate the stability profiles with observations. To address this concern, we have:**
  - **Added available temperature, wind speed, and wind direction observations for the FINO1 site in the relevant (now Fig. 4) figure**
  - **Added available temperature, potential temperature, wind speed, wind direction, and TKE observations for the aircraft transect site in the relevant (now Fig. 4) figure**

- Added available temperature, potential temperature, wind speed, wind direction, and TKE observations for the aircraft profiles in a new (now Fig. 5) figure

Because the turbines themselves do influence the local environment that we are analyzing, we also include a separate analysis of NWF cases in Sect. A1 of the Appendix.

[Figure]

**Figure 4. Temperature, potential temperature, wind direction, wind speed, and TKE vertical profiles from observations and WRF simulations for both sites. In all cases, the dashed lines indicate the modeled PBL height and the grey region indicates the turbine rotor region. Observations in the aircraft region are separated between and even and odd transects. (a) FINO1 temperature; (b) FINO1 potential temperature; (c) FINO1 wind direction; (d) FINO1 horizontal wind speed; (e) FINO1 TKE; (f) aircraft temperature; (g) aircraft potential temperature; (h) aircraft wind direction; (i) aircraft horizontal wind speed; (j) aircraft TKE. FINO1 cases are averaged over hours 12:00:00–00:00:00 and the aircraft region cases are averaged over 14:10:00–16:10:00. FINO1 TKE calculations based on observations were not available due to the coarse temporal resolution of the wind speeds. FINO1 potential temperature calculations based on observations were not available due to a lack of pressure observations.**

[Figure]

**Figure 5. Observed and modeled vertical profiles for the aircraft vertical profile flights (Table 3, Fig. 1b). In all cases, the horizontal line indicates the modeled PBL height, and the color differentiates the PBL scheme. The top row of panels corresponds to modeled output and the bottom row of panels corresponds to the aircraft profile observations. Modeled output are determined to be a given middle cell for each profile as in Larsén and Fischereit (2021) based on the timestep indicated in Table 3. (a) modeled temperature; (b) modeled potential temperature; (c) modeled wind direction; (d) modeled horizontal wind speed; (e) modeled TKE; (f) observed temperature; (g) observed potential temperature; (h) observed wind direction; (i) observed horizontal wind speed; (j) observed TKE.**

- *L. 244: Add PBLH as reference*

**Thank you for this comment. We have added PBLH to both our vertical profiles (Fig. 4 and Fig. 5) as shown above.**

- *Fig. 6: why are there only contour plots for TKE, and not for WS? Suggest to either add WS contour plots, or just add TKE profile to Fig. 5.*

Thank you for this thoughtful comment. We address this concern by adding a TKE profile to both the vertical profiles (Fig. 4 and Fig. 5) as shown above. We also introduce a WS contour plot (Fig. 7).

[Figure]

**Figure 7. Modeled wind speed cross-section at a constant latitude of 54.03. (a) FINO1 3fa_025; FINO1 mfa_025; (e) FINO1 3fa_025- mfa_025; (b) aircraft 3fa_025; (d) aircraft mfa_025; (f) aircraft 3fa_025 - mfa_025. The horizontal dashed black line denotes the average modeled PBL height, the star indicates the FINO1 tower location, the "X" marks the first transect path, and the black circles indicate the turbine hub height.**

- *Sect. 3.2.1: Why is there not TKE/TI comparison for FINO1?*

**Thank you for this careful observation. We have not added a TKE/TI comparison for FINO1 because of the coarse (~10 minute) temporal resolution. This decision to not include TKE/TI is also corroborated in the work of Ali et al. (2023), both in the main text and in their Appendix. This decision is noted on (our) line 185: "TKE calculations at the FINO1 site were not available due to coarse temporal resolution of the wind observations at the FINO1 site".**

- *L. 281-282: increasing TKE factor increases wind speed*

**We thank the reviewer for this careful observation. This sentence has been removed as the discussion of the optimal wind farm TKE factor has been moved to the Appendix.**

- *L. 287-288: what should be noted is that the wind speeds a lower altitudes are also lower, resulting in lower biases. Alternatively look at relative biases*

**We thank the reviewer for this observation. This observation is included in a discussion of (updated) Fig. 10 on (updated) line 441, which reads "[f]urther, the wind speeds at lower altitudes are also slower, resulting in smaller bias."**

- *Fig. 8: unclear what the mean values are*

**Thank you, we acknowledge the lack of clarity in this figure, now Fig. 9. Reviewer 2 has similarly expressed confusion about this plot. By focusing on only one wind farm TKE factor, we hope that this figure is now easier to interpret.**

[Figure]

**Figure 9. Time series of 76 m modeled horizontal wind speeds (WS) compared to 81 m FINO1 observations for the hours of 12:00:00–00:00:00. Both the modeled wind speeds and observed wind speeds are resampled to 30 minutes.**

- *Fig. 11: relate findings to FINO results at multiple heights: can these results be extrapolated to arrive at the results shown here? Additionally, 11c shows the opposite to FINO, please explain why.*

**Thank you for this observation. We believe that these two figures (now Fig. 10-FINO and Fig. 12-aircraft) capture different domains. While the FINO1 figure captures performance differences across multiple heights, the aircraft region figure captures performance differences across spatial transects at the same height. As such, while our main result, which is that the optimal PBL scheme depends on the site, is reinforced in this figure, further comparisons between these figures are not obvious. Our discussion of the differences in the optimal PBL scheme are best exemplified in our discussion section starting on new line 525:**

**"Differences in the relative measurement height between the two sites in this case study may also affect the PBL comparison. Whereas the FINO1 tower is within the turbine rotor region, the aircraft measurements are taken more than 100 m above the turbines in a stably stratified boundary layer (Fig. 4) that suppresses some interactions between the atmosphere sampled at the turbine level and the aircraft level (Fig. 6). Thus, the 3DPBL scheme improves cRMSE turbine-induced turbulence characterization in the turbine rotor region as sampled at FINO1 (Fig. 14d) and overpredicts turbine-induced turbulence aloft as sampled by the aircraft measurements, on average (Fig. 14c)"**

- L. 307: The issue of relative sensor alignment is very relevant and deserves more attention than this single sentence. Mention this in the methods section when introducing the aircraft measurements and include it in the discussion of uncertainties.

**We thank the reviewer for their thoughtful comment. We agree that a more detailed discussion of the uncertainties related to the aircraft data helps contextualize the results. We have expanded our discussion of the relative sensor alignment both in the methods section (Sect. 2.4.2) in which we introduce the aircraft data processing on line 311:**

**"These wind speed (and corresponding TKE) retrievals were subject to errors in aircraft yaw alignment. Bärfuss et al. (2023) analyzed experimental influences on retrievals from a drone platform and concluded yaw misalignment to be the dominant source of error for wind speed (and TKE) retrievals. Bärfuss et al. (2023) also noted that errors in yaw alignment have opposite signs depending on whether the sensor interacts with the wind from the starboard or backboard side. Because odd transects expose the sensor to one side of the plane and even transects expose the sensor to the other side, this error in yaw alignment could be reflected in systematic differences in wind speed (and TKE) between even transects and odd transects."**

as well as in the results section where we analyze measurement variability (Sect. 3.2.2) starting on line 458:

"This systematic transect variability corresponds to the reversed directions of the transect paths – whereas transects 1, 3, and 5 are performed in the northwesterly direction, transects 2, 4, and 6 are performed in the southeasterly direction. This directional variability could potentially be explained by errors in yaw alignment. As noted earlier, the dominant source of error in one set of wind speed retrievals were previously determined to be the yaw measurements (Bärfuss et al., 2023). The error in yaw alignment was also determined to have alternating signs on whether the wind approached the sensor from the starboard or the backboard side of the aircraft (Bärfuss et al., 2023). Because the even and odd transects involved opposite alignments of the aircraft, it would follow that the odd and even transects would then show distinct wind speed (and TKE) errors."

- *L. 355: Fig. 14a is aircraft, but text says FINO1.*

Thank you. We have updated this line, now starting on line 528, to reflect the appropriate sites with a new, condensed Fig. 14.

- *L. 363: Provide references for "recent scientific discussion"*

We thank the reviewer for this thoughtful comment. We stage our justification of the use of the wind farm TKE factor of 0.25 in several stages. This discussion, now starting on line 225, reads:

"Recent scientific discussion (Mangara et al., 2019; Archer et al., 2020; Siedersleben et al., 2020; Sanchez Gomez et al., 2023; Ali et al., 2023; García-Santiago et al., 2024; Optis, 2024) has focused on the determination of the optimal value of the wind farm TKE factor, $C_{TKE}$ , with varying conclusions…"

After discussing these works, we then finish the discussion by noting:

"We considered TKE factors of 0, 0.25, and 1. We focused on these three wind farm TKE factors both to cover the full range of variability and to consider the 0.25 factor suggested by Archer et al. (2020). We present the results with a 0.25 wind farm TKE factor here in the main text and discuss the results for the other wind farm TKE factors in Sect. A2."

- *L. 433-447: This outlook does not belong to the conclusion section.*

We thank the reviewer for this comment. We believe that analyzing data from other field campaigns is the most natural way to address the reviewer's expressed concern about

the ability to extend our conclusions to other sites. As such, we are hesitant to remove this discussion. Further, because the third Wind Farm Improvement Project (WFIP3) is still active, we believe that the most natural location for this discussion is in the conclusion section. We revise this discussion, which now starts on line 557 of the conclusion, to:

"Subsequent investigations could explore other case studies to provide perspective into the generalizability of the results across other sites. Similarly, datasets from the third Wind Forecast Improvement Project (WFIP3) could be useful to explore how offshore wind characterization might differ between the North Sea and the eastern United States (WFIP3). Moreover, datasets from the land-based, horizontally homogeneous American WAKE experimeNt – or AWAKEN – campaign (Moriarty et al., 2024) could be useful to study because previous land-based studies analyzing the 3DPBL scheme have involved complex terrain and far fewer detailed observations."

- *L. 461-462: If you claim this, show that the 3DPLB scheme indeed extracts more momentum for higher altitudes.*

We thank the reviewer for this thoughtful comment. To address this concern, we created an additional figure (Fig. 7) that shows wind speed vertical cross sections at each site and reflects the momentum extraction through reduced wind speeds. In Fig. 7f, we see that the 3DPBL-MYNN difference in wind speeds at the aircraft location (X) is stronger than below. We also note the influence of the additional mechanism such that the turbines extract momentum from aloft, increasing wind speeds. Starting on line 407, we state:

"These diverging patterns in TKE characterization between the two sites also have secondary influences on measured wind speeds (Fig. 7). The 3DPBL scheme enhances TKE peaks present with the MYNN scheme for both the aircraft region (Fig. 6b,d) and the FINO1 region (Fig. 6a,c). However, the differing measurement heights within the stable stratification capture these effects differently. In the aircraft region, the enhanced TKE implies greater momentum extraction and, consequently, reduced wind speeds at the measurement site (Fig. 7b,d,f). The rotor layer in the FINO1 region, however, introduces an additional mechanism. Here, TKE from aloft mixes more momentum from aloft into the measurement region, which actually increases wind speeds (Fig. 7a,c,e). Thus, the differing measurement altitudes between the two sites may impact both the TKE and wind speed assessments presented below."

[Figure]

**Figure 7. Modeled wind speed cross-section at a constant latitude of 54.03. (a) FINO1 3fa_025; FINO1 mfa_025; (e) FINO1 3fa_025- mfa_025; (b) aircraft 3fa_025; (d) aircraft mfa_025; (f) aircraft 3fa_025 - mfa_025. The horizontal dashed black line denotes the average modeled PBL height, the star indicates the FINO1 tower location, the "X" marks the first transect path, and the black circles indicate the turbine hub height.**

- *L. 509-523: Larsen & Fishereit is mentioned 7 times in just a few sentences. Rewrite this. This happens all over the manuscript, please review that you don't insert unnecessary references.*

Thank you, we have reworked the sentence structure of this paragraph in order to reduce the number of references. This paragraph now starts on line 245:

"As such, the results presented in Siedersleben et al. (2020) are qualitatively different from those presented in Larsén and Fischereit (2021). Although Larsén and Fischereit (2021) performed their analysis after the advection bug was addressed and performed simulations with advection both on and off, Larsén and Fischereit (2021) argue that further analysis would be necessary to make a formal recommendation."

- *Appendix A3: You show here that the results shown in the main body of the manuscript hold for just the 2h aircraft data is available. If this is true, why show the 12h analysis at all? Explain in the main body of the text why you argue that the 12h analysis is better. Additionally, FINO1 allows for an analysis at a much longer time scale. Justify why you restrict your analysis to just 12 hours.*

We thank the reviewer for their thoughtful comment. Reviewer 2 similarly addressed the duration of the analysis. The 2 hours of aircraft data aligned with the 12 hours of FINO1 analysis were strategically chosen to accomplish several goals, including:
- Contextualizing our results with the broader literature for this case study (i.e. consistency with Ali et al. (2023)
- Providing some level of indication as to the diurnal consistency of our results. For example, the optimal PBL scheme in the FINO1 region is consistent throughout the time period of the study
- Restricting our analysis to high-quality, in situ observations without introducing additional uncertainties associated with methods like statistical downscaling
- Restricting our analysis to stably-stratified conditions, which contribute the strongest and longest wakes.

Thus, by restricting our analysis to conditions that are well-suited for wake analysis, with in situ observations that reduce uncertainty, and also also allow us to contextualize our results with the broader literature for this case study, we believe our analysis reframes the apparent lack of available long term in situ offshore wind farm observations into a well-suited case study analysis for documenting physical differences between models.

We also now include a similar justification in our discussion section in (updated) lines 531-536:

"The stable stratification present in this case study also improves the utility of the results of this PBL comparison. By restricting this analysis to time periods considered in previous analyses for this case study, not only are the results contextualized within the broader literature, but the conditions that contribute the strongest and longest wakes are also highlighted. Thus, while other analyses of this region may approach the lack of available in situ observations by introducing statistical downscaling methods to explore scientific questions around diurnal, seasonal, and climatic trends (Fischereit et al.,

**2022b), this analysis instead addresses scientific questions that are best-suited with in situ observations alone.”**

**References**

Ali, K., Schultz, D. M., Revell, A., Stallard, T., & Ouro, P. (2023). Assessment of five wind-farm parameterizations in the Weather Research and Forecasting model: A case study of wind farms in the North Sea. *Monthly Weather Review*, *1*(aop). https://doi.org/10.1175/MWR-D-23-0006.1

Arthur, R. S., Rybchuk, A., Juliano, T. W., Rios, G., Wharton, S., Lundquist, J. K., & Fast, J. D. (2024). Evaluating mesoscale model predictions of diurnal speedup events in the Altamont Pass Wind Resource Area of California. *Wind Energy Science Discussions*, 1–30. https://doi.org/10.5194/wes-2024-137

Bärfuss, K. B., Schmithüsen, H., & Lampert, A. (2023). Drone-based meteorological observations up to the tropopause – a concept study. *Atmospheric Measurement Techniques*, *16*(15), 3739–3765. https://doi.org/10.5194/amt-16-3739-2023

Fischereit, J., Brown, R., Larsén, X. G., Badger, J., & Hawkes, G. (2022). Review of Mesoscale Wind-Farm Parametrizations and Their Applications. *Boundary-Layer Meteorology*, *182*(2), 175–224. https://doi.org/10.1007/s10546-021-00652-y

Larsén, X. G., & Fischereit, J. (2021). A case study of wind farm effects using two wake parameterizations in the Weather Research and Forecasting (WRF) model (V3.7.1) in the presence of low-level jets. *Geoscientific Model Development*, *14*(6), 3141–3158. https://doi.org/10.5194/gmd-14-3141-2021

Peña, A., García-Santiago, O., Kosović, B., Mirocha, J. D., & Juliano, T. W. (2023). Can we yet do a fairer and more complete validation of wind farm parametrizations in the WRF model? *Journal of Physics: Conference Series*, *2505*(1), 012024. https://doi.org/10.1088/1742-6596/2505/1/012024

Platis, A., Siedersleben, S. K., Bange, J., Lampert, A., Bärfuss, K., Hankers, R., et al. (2018).

First in situ evidence of wakes in the far field behind offshore wind farms. *Scientific Reports*, *8*(1), 2163. https://doi.org/10.1038/s41598-018-20389-y

Rybchuk, A., Juliano, T. W., Lundquist, J. K., Rosencrans, D., Bodini, N., & Optis, M. (2022).

The sensitivity of the Fitch wind farm parameterization to a three-dimensional planetary boundary layer scheme. *Wind Energy Science*, *7*(5), 2085–2098. https://doi.org/10.5194/wes-7-2085-2022

Siedersleben, S. K., Platis, A., Lundquist, J. K., Djath, B., Lampert, A., Bärfuss, K., et al. (2020).

Turbulent kinetic energy over large offshore wind farms observed and simulated by the mesoscale model WRF (3.8.1). *Geoscientific Model Development*, *13*(1), 249–268. https://doi.org/10.5194/gmd-13-249-2020

---

## Author Comment (AC2)

* * *
*Editor's and reviewers' comments appear in italics*; **our responses appear in boldface blue text**.

**Reviewer 2:**

*The study evaluates two planetary boundary layer (PBL) schemes in WRF in the North Sea. Both are compatible with the Fitch wind farm parameterization. For validation, high-resolution measurements of a research aircraft and the FINO1 mast measurements are used. The study finds that the optimal PBL scheme varies with the location, quantity of interest, and error metric.*

*In contrast to what the title suggests, not only the PBL scheme is varied, but also the TKE advection within WRF is activated and deactivated, and the sensitivity of the results to the TKE factor within the WFP by Fitch et al. is investigated. Due to the many different aspects being studied, as well as the two parameters, different error matrices, and two locations, the results section is rather clouded, and it is difficult to extract the main message. On the other hand, it shows how complex the situation is and that a simple answer to the question of the "optimal" scheme is impossible to give. I think this in itself is an important conclusion and therefore I recommend publishing this article, after some revisions.*

**We thank the reviewer for their thoughtful assessment and appreciation of the complexity of an "optimal assessment." We believe that the above appropriately characterizes the scope of our analysis.**

*Major comments*

*(1) The authors put a lot of effort into investigating this single case study of (mainly) 2 hours (although some FINO1 analysis stretches over the period of 1 day). This imminently raises the question of how transferable those results are to other meteorological conditions (and sites). While the conclusions state that "Subsequent investigations could explore other case studies to provide perspective into the generalizability of the results across other sites." the author also critically highlights from previous literature that "Conclusions drawn from these validation studies may also be influenced by site-specific or meteorological conditions." Thus, the discussion on the generalizability should be extended to highlight why it is worth spending so much effort on just 2 hours.*

We thank the reviewer for their thoughtful comment. Reviewer 1 similarly addressed the duration of the analysis. The 2 hours of aircraft data aligned with the 12 hours of FINO1 analysis were strategically chosen to accomplish several goals, including:

- Contextualizing our results with the broader literature for this case study (i.e. consistency with Ali et al. (2023)
- Providing some level of indication as to the diurnal consistency of our results. For example, the optimal PBL scheme in the FINO1 region is consistent throughout the time period of the study
- Restricting our analysis to high-quality, in situ observations without introducing additional uncertainties associated with methods like statistical downscaling
- Restricting our analysis to stably-stratified conditions, which contribute the strongest and longest wakes.

Thus, by restricting our analysis to conditions that are well-suited for wake analysis, with in situ observations that reduce uncertainty, and also also allow us to contextualize our results with the broader literature for this case study, we believe our analysis reframes the apparent lack of available long term in situ offshore wind farm observations into a well-suited case study analysis for documenting physical differences between models.

We also now include a similar justification in our discussion section in (updated) lines 531-536:

"The stable stratification present in this case study also improves the utility of the results of this PBL comparison. By restricting this analysis to time periods considered in previous analyses for this case study, not only are the results contextualized within the broader literature, but the conditions that contribute the strongest and longest wakes are also highlighted. Thus, while other analyses of this region may approach the lack of available in situ observations by introducing statistical downscaling methods to explore scientific questions around diurnal, seasonal, and climatic trends (Fischereit et al., 2022b), this analysis instead addresses scientific questions that are best-suited with in situ observations alone."

*(2) The study mainly compares the two PBL schemes, the 3DPBL and the MYNN scheme, in a wind farm context. Thus, along with a short introduction to the WFP by Fitch et al., it should also provide some introduction to the two schemes and the difference between those to extend the too brief description that is given in the introduction (line 72-81). This will also aid the interpretation of the results.*

We thank the reviewer for this thoughtful comment. We have added a full subsection to our methods section (the new Sect. 2.1) that includes the governing equations for both models and discusses the key differences between the PBL schemes. We then reference this section throughout the manuscript to aid interpreting the physical differences between the schemes. As some examples:

- **L. 351: "MYNN simulations … the surface."**

In this section, now starting on line 351, we argue that the slight differences in stability are a consequence of the increased TKE from the 3DPBL scheme. This increased TKE with the 3DPBL scheme encourages slightly more mixing:

"Because the modeled air temperatures are almost identical between models (Fig. 4a,f), these slight differences in the surface stability could be a consequence of the greater TKE with the 3DPBL scheme that encourages slightly more mixing"

- **L. 390: "Both the … simulation (Fig. 6a,c)."**

In this section, now starting on line 390, we point out that the larger TKE with the 3DPBL scheme in stable conditions, which is consistent with the trend identified under the idealized, stable conditions simulated in Rybchuk et al. (2022), reflects the fundamental differences between the 3DPBL and MYNN models:

"These differences in TKE between the PBL schemes, which are consistent with those identified under the idealized, stable conditions simulated in Rybchuk et al. (2022), reflect the fundamental differences between the models. Notably, while the MYNN scheme uses Smagorinsky mixing to characterize horizontal turbulent mixing, the 3DPBL scheme instead calculates the horizontal turbulent flux divergences explicitly. The two models also rely on different length scales and empirical constants"

- **L. 417 "MYNN average … wind speeds."**

In this section, now starting on line 417, we maintain our interpretation that larger TKE with the 3DPBL scheme implies greater momentum extraction that results in reduced wind speeds.

"This larger TKE with the 3DPBL scheme extracts more momentum from the mean wind, resulting in a greater reduction in wind speed. This finding that MYNN wind speeds are faster than 3DPBL wind speeds is consistent with other comparisons of these two PBL schemes, completed in both real and idealized conditions (Juliano et al., 2022; Rybchuk et al., 2022; Arthur et al., 2022; Peña et al., 2023; Arthur et al., 2024)."

We also introduce a new Fig. 7 to document the differences in vertical structure of wind speeds between the two PBL schemes to support our claim.

[Figure]

**Figure 7. Modeled wind speed cross-section at a constant latitude of 54.03. (a) FINO1 3fa_025; FINO1 mfa_025; (e) FINO1 3fa_025- mfa_025; (b) aircraft 3fa_025; (d) aircraft mfa_025; (f) aircraft 3fa_025 - mfa_025. The horizontal dashed black line denotes the average modeled PBL height, the star indicates the FINO1 tower location, the "X" marks the first transect path, and the black circles indicate the turbine hub height.**

**Finally, we corroborate this interpretation in Sect. A2 of the Appendix. Notably, Fig. A3a of Sect. A2 demonstrates how increasing the amount of TKE (in this case, via the wind farm TKE factor) reduces the wind speeds.**

[Figure]

**Figure A3.** Error metric box plot for aircraft observations collected at 250 m for wind speed. The box and whiskers describe aircraft transect variability and are based on Q1 (25th percentile), Q3 (75th percentile), and the interquartile range (IQR) (Q3–Q1). The box encloses the IQR, and the whiskers extend to Q1-1.5*IQR and Q3+1.5*IQR. The simulation names are mapped according to the short names provided in Table 6, and the vertical dotted lines visually separate simulations by wind farm TKE factor. (a) Wind speed bias; (b) wind speed cRMSE; (c) wind speed R2; (d) wind speed EMD.

- L. 449 "TKE in … MYNN simulations."

**In this section, now starting on line 449, we reference the figure from the discussion in the previous bullet point and maintain the same interpretation.**

**"As noted earlier, these differences in TKE between the PBL schemes reflect the fundamental differences between the models in length scales, empirical constants, and horizontal mixing approaches."**

*(3) Validation of atmospheric stability: Stability is an important parameter for analyzing the results. However, while it is discussed thoroughly from the simulations, the profiles are not validated with the measurements in the lower part of the atmosphere. Both the FINO measurements and the profiles flown by the aircraft could be used to evaluate the simulations in the lower part of the PBL.*

**We thank the reviewer for their thoughtful guidance. We now include vertical profiles both at the FINO1 tower and from the aircraft profile flights to demonstrate the stable stratification in the new Fig. 4 and Fig. 5. The stable profiles that we now provide in this manuscript are also corroborated by those in Larsén & Fischereit(2021) and Ali et al. (2023).**

[Figure]

**Figure 4. Temperature, potential temperature, wind direction, wind speed, and TKE vertical profiles from observations and WRF simulations for both sites. In all cases, the dashed lines indicate the modeled PBL height and the grey region indicates the turbine rotor region. Observations in the aircraft region are separated between and even and odd transects. (a) FINO1 temperature; (b) FINO1 potential temperature; (c) FINO1 wind direction; (d) FINO1 horizontal wind speed; (e) FINO1 TKE; (f) aircraft temperature; (g) aircraft potential temperature; (h) aircraft wind direction; (i) aircraft horizontal wind speed; (j) aircraft TKE. FINO1 cases are averaged over hours 12:00:00–00:00:00 and the aircraft region cases are averaged over 14:10:00–16:10:00. FINO1 TKE calculations based on observations were not available due to the coarse temporal resolution of the wind speeds. FINO1 potential temperature calculations based on observations were not available due to a lack of pressure observations.**

[Figure]

**Figure 5. Observed and modeled vertical profiles for the aircraft vertical profile flights (Table 3, Fig. 1b). In all cases, the horizontal line indicates the modeled PBL height, and the color differentiates the PBL scheme. The top row of panels corresponds to modeled output and the bottom row of panels corresponds to the aircraft profile observations. Modeled output are determined to be a given middle cell for each profile as in Larsén and Fischereit (2021) based on the timestep indicated in Table 3. (a) modeled temperature; (b) modeled potential temperature; (c) modeled wind direction; (d) modeled horizontal wind speed; (e) modeled TKE; (f) observed temperature; (g) observed potential temperature; (h) observed wind direction; (i) observed horizontal wind speed; (j) observed TKE.**

*(4) The appendix is too long and not only adds supporting information but presents new findings, e.g., regarding the effect of TKE advection. I suggest moving the section on the TKE advection to the main text, while some plots can probably be kept in the appendix (or removed completely) to limit the size of the main text. Furthermore, some additional figures can be removed if they do not add any new information. This would make the Appendix much more accessible, as now it is flooded with figures.*

We thank the reviewer for this thoughtful guidance, which is also echoed by Reviewer 1. Based on this joint guidance, we have revised the Appendix in the following method to make it more of a story by doing the following:

- We have removed the entirety of Appendix B from the text
- We have preserved the NWF vs. WF distinction as part of the appendix to better distinguish the differences between PBL schemes in "baseline" conditions (Sect A1)
- We have moved the discussion of the effect of the wind farm TKE factor to the Appendix (Sect A2)
- We have kept the advection section in the Appendix (now Sect A3) to focus the manuscript. We considered the reviewer's suggestion to move advection into the main text but we feel that would unnecessarily broaden the discussion in the main paper.

*Specific comments*

*- Line 47: Other (newer) references should be mentioned*

We thank the reviewer for this comment. Reviewer 1 similarly acknowledged that a citation from 2014 was not necessarily representative of the most current advancements in wind farm parameterization validations. We have updated this reference to a more recent, highly-cited review of wind farm parameterization validations (Fischereit et al., 2022).

*- Line 59: Add citation for EWP*

Thank you, we have added the citation for EWP on (updated) line 62.

*- Table 1: "Select" -> "Selected"*

Thank you, we have adjusted the Table 1 caption to show "selected" instead of "Select".

*- Section 2.2 Provide a tabular overview of the WRF settings to provide a better overview for the readers. Also, consider publishing the namelist settings on Zenodo for reproducibility.*

Thank you, we have added a table of select WRF settings (Table 5) and also intend to publish our namelist settings on Zenodo upon acceptance of the paper as noted in the Data Availability section.

*- Line 131: Which WRF version is used precisely?*

We have added a reference to WRFV4.4.2 on line 206.

*- Line 157: Consider repeating, "The results from the other runs are analyzed in the Appendix."*

**Thank you, we repeatedly emphasize that the results of the other runs are analyzed in the Appendix on (updated) lines 224, 240, and 254.**

*- Line 205: Consider removing "also"*

**Thank you, we have removed "also" on updated line 319.**

*- Figure 4: Why are not all curves labeled? This might be useful for other researchers.*

**Thank you, we have added labels and unique colors to each of the drag proxy curves for what is now Fig. 3.**

[Figure]

**Figure 3. (a) Curve illustrating turbine $C_T$, $C_P$, and power specifications for the turbine model in the Gode wind farm. (b) Drag proxy for each of the eight turbine models present in this case study.**

*- Figure 8: Why is the Mean (solid line) sometimes above the single realizations (as shown by the symbols)? Or does "Mean" mean something else here?*

**Thank you, we acknowledge the lack of clarity in this figure (now Fig. 9). Reviewer 1 has similarly expressed confusion about this plot. By focusing on only one wind farm TKE factor, we hope that this figure is now easier to interpret.**

[Figure]

**Figure 9. Time series of 76 m modeled horizontal wind speeds (WS) compared to 81 m FINO1 observations for the hours of 12:00:00–00:00:00. Both the modeled wind speeds and observed wind speeds are resampled to 30 minutes.**

*- Figure 14 / 15 captions: Put the analyzed quantity at the beginning of the caption to easier see the difference between those two figures:  Error metric box plot for aircraft observations collected at 250 m for wind speed / TKE.*

**Thank you, we have adjusted our captions to read as "Error metric box plot for aircraft observations collected at 250 m for wind speed / TKE." These figures are now listed as Fig. A3 and Fig. A4.**

*- Line 408: add space between "(2023)," and "Optis"*

**Thank you, we agree that there should have been a space. However, this sentence is now removed as part of the removal of the discussion of the optimal wind farm TKE factor.**

*- Figure A1 caption: Use also abbreviations as documented in table 4 in the caption*

**Thank you, we have introduced the Table 4 abbreviations in the caption for Fig. A1.**

*- Figure B10-B13: See comment regarding figure 14/15*

**Thank you, this figure has been removed entirely, consistent with the distilling of the Appendix.**

*- Line 632: "..wind speeds - to - outperform ... " add "to" here?*

**Thank you, we agree. This sentence has been removed entirely, consistent with the distilling of the Appendix.**

**References**

Ali, K., Schultz, D. M., Revell, A., Stallard, T., & Ouro, P. (2023). Assessment of five wind-farm parameterizations in the Weather Research and Forecasting model: A case study of wind farms in the North Sea. *Monthly Weather Review*, *1*(aop). https://doi.org/10.1175/MWR-D-23-0006.1

Bärfuss, K. B., Schmithüsen, H., & Lampert, A. (2023). Drone-based meteorological observations up to the tropopause – a concept study. *Atmospheric Measurement Techniques*, *16*(15), 3739–3765. https://doi.org/10.5194/amt-16-3739-2023

Fischereit, J., Brown, R., Larsén, X. G., Badger, J., & Hawkes, G. (2022). Review of Mesoscale Wind-Farm Parametrizations and Their Applications. *Boundary-Layer Meteorology*, *182*(2), 175–224. https://doi.org/10.1007/s10546-021-00652-y

Larsén, X. G., & Fischereit, J. (2021). A case study of wind farm effects using two wake parameterizations in the Weather Research and Forecasting (WRF) model (V3.7.1) in the presence of low-level jets. *Geoscientific Model Development*, *14*(6), 3141–3158. https://doi.org/10.5194/gmd-14-3141-2021

Rybchuk, A., Juliano, T. W., Lundquist, J. K., Rosencrans, D., Bodini, N., & Optis, M. (2022). The sensitivity of the Fitch wind farm parameterization to a three-dimensional planetary boundary layer scheme. *Wind Energy Science*, *7*(5), 2085–2098. https://doi.org/10.5194/wes-7-2085-2022